# Continuous dynamics of cooperation and competition in social decision-making
Darius Lewen [1,2,4], Vladyslav Ivanov [3,4], Jonas Dehning [1,2,4], Johannes Ruß[2], Anna Fischer[2], Lars Penke[2], Anne Schacht [2], Alexander Gail[2,3], Viola Priesemann [1,2,5] ✉ & Igor Kagan [3,5] ✉

Real-life social interactions often unfold continuously and involve dynamic cooperation and competition, yet most studies rely on discrete games that do not capture the adaptive and graded nature of continuous sensorimotor decisions. To address this gap, we developed the Cooperation-Competition Foraging game—an ecologically grounded paradigm in which pairs of participants (dyads) navigate a continuous shared space under face-to-face visibility, deciding in real-time to collect rewarded targets either individually or jointly. Dyads ($n = 58$, 116 participants) spontaneously converged on distinct stable strategies along the cooperation-competition spectrum, forming three groups: cooperative, intermediate, and competitive. Despite the behavioral complexity, our computational model, which incorporated travel path minimization, sensorimotor communication, and recent choice history, predicted dyadic decisions with 87% accuracy, and linked prediction certainty with ensuing dynamics of spatiotemporal coordination. Further modeling revealed how sensorimotor factors, such as movement speed and skill, shape distinct strategies and payoffs. Crucially, we quantify the cost of cooperation, demonstrating that in many dyads prosocial tendencies outweigh the individual benefits of exploiting skill advantages. Our versatile framework provides a predictive, mechanistic account of how social and embodied drivers promote the emergence of dynamic cooperation and competition, and offers rigorous metrics for investigating the neural basis of naturalistic social interactions, and for linking personality traits to distinct strategies.

People and animals often face choices that require balancing or alternating between cooperative and competitive strategies, especially in settings involving shared or limited resources. Whether foraging for food, hunting, or engaging in economic exchanges, the dynamics of cooperation and competition play a crucial role in determining outcomes such as efficiency, fairness, and overall success. In *dyadic* interactions between two individuals, each must evaluate not only their own goals and actions but also the intentions and actions of their partner. However, it is often uncertain whether others intend to compete or cooperate, and how to flexibly adjust strategies in such volatile situations[1]. Furthermore, these interactions often unfold continuously in time and space, grounded in real-time, embodied decision-making. Understanding how such decisions are made, and what drives cooperation or competition, is essential for unraveling the fundamentals of social behavior[2–5]. Here, we elucidate the behavioral and computational mechanisms of continuous, embodied dyadic foraging, using an innovative paradigm where human

participants can freely choose cooperative, competitive, and intermediate strategies.

Classical game-theoretical approaches have relied on discrete strategic interactions, focusing on a binary dichotomy between cooperation and competition[6–12]. While some resource allocation paradigms, such as the Ultimatum, Dictator, Trust and Public Goods games, involve gradual options (e.g., how much to share)[13,14], typical 2 × 2 matrix games use binary choices. Competitive zero-sum games and pure coordination games allow for only one meaningful strategy. Even in mixed-motive dilemmas such as Prisoner's Dilemma, Stag Hunt, and Chicken / Hawk-Dove, decisions are reduced to a stereotyped binary choice between cooperation and selfish defection. Yet, real-world decisions often transcend such dichotomies[15]. To enable a continuous spectrum of strategies that reflect the non-dichotomous nature of social decisions, a recent study developed a game called the Space Dilemma, where pairs of participants are presented with a spatial choice on a continuous one-dimensional scale between cooperation and competition[16].

[1]Max Planck Institute for Dynamics and Self-Organization, Am Faßberg 17, Göttingen, Germany. [2]University of Göttingen, Wilhelmsplatz 1, Göttingen, Germany. [3]German Primate Center – Leibniz Institute for Primate Research, Kellnerweg 4, Göttingen, Germany. [4]These authors contributed equally: Darius Lewen, Vladyslav Ivanov, Jonas Dehning. [5]These authors jointly supervised this work: Viola Priesemann, Igor Kagan. ✉e-mail: viola.priesemann@ds.mpg.de; ikagan@dpz.eu

The continuous nature of this game is however limited to a single choice in each discrete trial, where each participant makes an isolated decision informed by the history of previous outcomes and predictions about their partner's upcoming decisions.

To capture the continuous complexity of realistic interactions—in both time and space—less structured and more dynamic pacing and flow are needed. Unlike discrete "simultaneous" and sequential turn-based games, natural social exchanges typically rely not only on predictions from past experience, but also on real-time, moment-to-moment information[17–19]. In animals, simple forms of associative learning are compromised when decisions are temporally separated from their consequences[20], and even in cognitively advanced nonhuman primates, coordination based on mutual choice history is more demanding than relying on the immediately observable actions of others[21–25]. Theoretical simulations show that cooperative populations evolve more easily under a continuous flow of information between agents[19], and that action visibility enhances cooperation in coordination games[26]. Likewise, transitioning from discrete to continuous repeated interactions promotes greater cooperation in humans[27,28], and synchronous action fosters cooperative economic exchanges[29]. Action visibility and movement timing may play an equally significant role in shaping competitive strategies[24,30–32]. More generally, individual dispositions and psychological traits lead to biases towards being more cooperative or more competitive, independent of the payoff formulation[33]. These biases might be especially prominent in continuous environments, where face-to-face and action visibility strongly influence strategic considerations[24,34–38]. These findings highlight the insights and possibilities gained by shifting from discrete to continuous decisions[39].

Importantly, the transition to continuous interactions not only shapes strategies; it also enables ethologically grounded, fundamental link between decision-making and sensorimotor control[40]. Unlike the classical serial models in which decisions precede actions, in *continuous action spaces*, choices are not solely driven by abstract computations of expected payoffs, but are interwoven with concurrent perceptual and motor processes[41–44], reflecting naturalistic "decide-while-acting" scenarios[45–49]. The dynamic feedback loops between observation and response allow adjusting decisions based on environmental cues and the actions of others, facilitating coordination and modulating cooperative or competitive inclinations depending on moment-to-moment evidence. Such embodied decision-making is especially relevant in foraging, where real-world constraints like effort expenditure and biomechanical limitations[50–52] impose cost-benefit trade-offs[53,54]. Foraging efficiency is shaped by the dynamic balance between cognitive effort, physical effort, and reward, along with continuous adjustments to feedback through perception-action coupling[55–58]. As a result, decisions in spatially and temporally rich environments emerge from a combination of deliberate strategies and sensorimotor-driven adjustments[59,60].

Here, we explore the interplay between reward, effort, cooperation and competition, focusing on how strategies emerge and are maintained during economic decision-making in a continuous dyadic foraging context. We developed a paradigm called "Cooperation–Competition Foraging" game that affords a wide range of interactions between two participants who can collect rewards together or independently. In line with increasingly common use of engaging game-like paradigms to study cognition[61,62], we designed a free-flowing social game, in which participants use continuous arm movements to select targets in a real-time face-to-face "transparent" setting[24,26,63], enhanced by compelling audiovisual feedback. By examining the spatiotemporal trajectories and strategic choices made by participants, we seek to capture basic cognitive processes that underpin these dynamic decisions.

The game features two types of targets: joint targets, which require coordinated cooperation for shared reward, and single targets, which can be collected by either participant. We hypothesized that this structure would engender a broad range of strategic interactions. On the one hand, the "gamified" environment with real monetary incentives could stimulate competitive tendencies, resulting in strategic positioning, race-like

dynamics, and influence of individual motor skills. Conversely, the immediacy of face-to-face engagement and continuous game flow could promote the inherent prosocial cooperativeness and propensity for fairness that characterizes many human interactions[64], leading to leader-follower dynamics and reciprocal turn-taking[24]. We anticipated that past dependencies and information flow between participants, such as cues that signal willingness to cooperate, would be higher during cooperative behaviors to facilitate coordination, compared to competitive interactions.

## Methods
### Participants
124 adult human participants participated in the study as paid volunteers. All participants gave written informed consent for participation after the procedures had been explained to them and before taking part in the experiment. Experiments were performed in accordance with institutional guidelines for experiments with humans and adhered to the principles of the Declaration of Helsinki. The experimental protocol was approved by the ethics committee of the Georg-Elias-Mueller-Institute for Psychology, University of Goettingen (GEMI 17-06-06 171). Participants were tested in pairs as 62 unique dyads, i.e., each participant contributed only once. 4 dyads were excluded from the current analysis because they exhibited large differences in behavior between blocks. The analyzed dataset included 116 participants in 58 unique dyads (mean ± SD age: 25 ± 4 years, range 18–36 years; 19 females and 97 males, as reported by participants; resulting in 46 male dyads, 7 female dyads, and 5 mixed female/male dyads). Most participants were male because they were recruited for a related ongoing study of male hormonal effects. These participants responded affirmatively to the question "Bist du chromosomal geschlechtlich männlich?" ("Are you chromosomally male?").

We did not collect or report data on participants' race, ethnicity, or other socially defined groupings, as these variables were not pertinent to the research questions addressed in this study.

### Experimental procedures
Pairs of participants (dyads) played the Cooperation–Competition Foraging game, sitting face-to-face across the table (120−140 cm inter-subject distance) with a large transparent screen in between (Eyevis 55 inch OLED, 1920 × 1080 pixels, 60 Hz refresh rate, Supplementary Movie S1, YouTube, OSF)[24,63]. The visual stimuli presented on the screen were visible from both sides. The task was implemented in Python 3.10 and run on Ubuntu 20.04 LTS. Prior to the experiment, participants received written instructions detailing the game mechanics, the payoff structure for each target type, and the procedure for determining their earnings (see Supplementary Information, Instructions for participants). Verbal clarifications were provided as needed to ensure full comprehension. In particular, participants were explicitly informed that their earnings would be performance-based, reflecting their cumulative payoffs during one randomly selected game block. Each experimental session consisted of two game blocks, each lasting 20 min. Participants were given breaks between blocks to minimize fatigue. Participants were not allowed to talk. At the end of the session, participants rolled a die to randomly determine which block's accumulated payoff would serve as their actual earnings.

There was no preregistration for this study.

### The cooperation–competition foraging game
The participants' objective in the dyadic Cooperation–Competition Foraging game is to earn money by collecting targets. To collect a target, participants were required to hover with their mouse-controlled cursors ("agents", blue and orange smaller circles, 2 cm diameter, 1.9 degrees of visual angle [°]) at 60 cm viewing distance) over the selected target (a bigger circle, 5 cm diameter, 4.8°) for one second. At any given time, the game field (a square with 51 cm side, 56°, with visible borders) contained three targets: one single target and two joint targets. All targets and agents were visible to both participants and were positioned randomly at the start of the session block using the 2D uniform distribution. After target collection (end of

collection cycle), the target of the same type immediately reappeared at a new random position, with no restrictions on reappearing near the previous position, but without overlap with the other two targets. The positions of both agents and the two remaining uncollected targets were not reset, ensuring a continuous transition between successive collection cycles.

The single target, which was white, could be collected by one participant alone ("winner-takes-all"). When an agent entered a single target, its color changed to match the agent's color, signaling whose agent was first and which participant would receive the payoff of 7 cent. The other participant received no payoff. Joint targets, which were partly blue, partly orange, required both participants to hover their agents over the target simultaneously to initiate the collection period. The color of joint target sectors reflected the asymmetric payoff distribution: one type offered 5 cent to the blue agent and 2 cent to the orange agent, and vice versa. The accumulated payoff of each participant was continuously shown in the lower right corner of the game field.

During target collection, a transparent disk expanded from the center to the edge of the target, visually indicating collection progress. Auditory feedback was provided through a sound with a continuously increasing pitch. If a participant left the target before the collection was complete, the progress was reset, and an error sound was played. Successful collection was confirmed by the target's disappearance and a short reward-associated sound.

To reduce the dependence of motor skill and to reflect the spatiotemporal limitations of realistic foraging, a maximum agent speed was set to to 42.6 cm/s. If a participant exceeded this maximum, their agent lagged relative to the mouse input. To ensure high temporal precision of the movement data, the agents' position was tracked at a sampling rate of 120 Hz.

## Timecourse of strategies over an experimental session

For the analysis of the stationarity of the strategies over an experimental session, we calculated the moving average of the fraction of single targets (FST) collected by both agents in dyad in a moving window of 1 min. For the analysis of the target choice prediction by different models, we used the moving average window of 30 s.

## Generalized Linear Model (GLM) for dyadic target choice

To model the choice behavior in the game, we analyzed which factors determine the target choice of the dyad in each collection cycle. Specifically, we predicted the collected target identity from the position of the two agents at the beginning of the collection cycle, the position of the targets, and potentially from the outcomes of the previous cycles. For distance-based predictions, the predicted target $j_{pred}$ is the one with the weighted minimal potential distance:

$$j_{pred} = \operatorname{argmin}_j \left[ D_S(1 - w), D_{J,A} \cdot w, D_{J,B} \cdot w \right]_j, \qquad (1)$$

where $D_S$ is the minimal distance from the single target to either agent, $D_{J,A}$ (or $D_{J,B}$) is the maximal distance between the joint target benefiting agent $A$ (or $B$) to either agent and $w$ a weighting factor such that the observed fraction of single targets (FST) is reached (for details, see Supplementary Methods 1.1).

This model was also used to simulate the optimal strategy for the dyad. For the range of possible weighting factors, we ran simulations of the game, where the choice of the target was given by the weighted distance formula above. We ran two different simulations: the agents either moved simultaneously or, when one agent was collecting a single target, the non-collecting agent placed itself on the game field such that the subsequent expected acquisition time is minimal (called "advantageous placement", see Supplementary Methods 1.1).

For the full GLM, we framed our model as a multiclass classification problem. We predicted the probability of observing the collection of each of the three targets given the vector $\vec{x}$, that is the distances to the targets, the identities of targets collected in previous cycles, and whether an "invitation"

is present in the current cycle (the non-collecting agent is near a joint target):

$$P(j_{obs} = j | \vec{x}) = \operatorname{Softmax} \left( \theta^T \cdot \vec{x} \right)_j. \qquad (2)$$

We took certain symmetries of the problems into account, mainly that the identity of both players are interchangeable, to reduce the number of parameters to be estimated (see Supplementary Methods 1.2). The regression coefficients $\theta$ were estimated by minimizing the loss with the Broyden-Fletcher-Goldfarb-Shanno (BFGS) algorithm. To avoid overfitting and for estimating the accuracy of the model, we used 5-fold cross-validation.

To quantify the uncertainty of dyad's choice, we measured the entropy $H$ of the target prediction:

$$H = - \sum_{j=1}^{3} P(j_{obs} = j | \vec{x}) \cdot \log_2 \left( P(j_{obs} = j | \vec{x}) \right). \qquad (3)$$

## Trajectory classification

To characterize the ongoing decision processes during the acquisition period, we classified spatiotemporal movement trajectories based on their shape relative to the targets, into six categories—"invitation", "failed invitation", "strongly curved", "different targets", "one ahead" and "concurrent"—using a set of heuristic rules for each category applied in that specific order. If the trajectory followed the rule, the trajectory was classified as such, if not, the next rule was tried. For precise definitions, see Supplementary Methods 1.3.

## Disentangling spatiotemporal factors shaping the payoff

To understand how trajectory efficiency, speed, and positioning influence agents' payoffs, we disentangled these different factors. We expressed the payoff as an addition/subtraction of the different factors that determine the trajectory length, divided by the mean speed, to obtain the contribution of each factor to the total joint payoff for each dyad. This decomposition is not exact: estimating the average payoff per cycle using the average trajectory length and mean speed introduces a small error, since the average of a ratio is not strictly equal to the ratio of averages. However, in this case, the correlation is nearly perfect ($r = 0.99$, see also Supplementary Methods 1.4).

## Competitive skill difference estimation

The skill difference between participants influences how successfully an agent competes for single targets. To estimate this competitive skill difference, we measured the proportion of contested single targets each agent managed to collect, focusing specifically on cycles where both agents attempted to reach the single target ("one ahead to the same target" and "concurrent to the same target" classes; see Supplementary Methods 1.5 and 1.6). Thus, we could not estimate the skill difference if the agents always played cooperatively using only the joint targets.

## Estimating of the cost of cooperation from counterfactual scenarios

To obtain an estimate of the cost of cooperation, i.e. an estimation of what would have happened if dyads had played more competitively than observed, we built a model that allows us to counterfactually estimate the payoff for other values of the FST $\Phi$ than observed for the dyad in question. Specifically, we estimated the payoff $\hat{R}^X(\Phi)$ of each agent $X$ by combining an estimate of the average joint payoff $\hat{R}(\Phi)$ with an estimate of the difference of payoff between the two agents in a dyad $\hat{R}_\Delta^X(\Phi)$ (see Supplementary Methods 1.6). The estimate of the average joint payoff was obtained by fitting the variables that shape the payoff, the average distance between target collections, the average reduction of this distance due to an advantageous placement, the average increase of distance due to trajectory curvature, and the average agent speed over all dyads. The difference of payoff between agents was estimated per dyad individually and is dependent on the skill difference of the participants. Thus, with

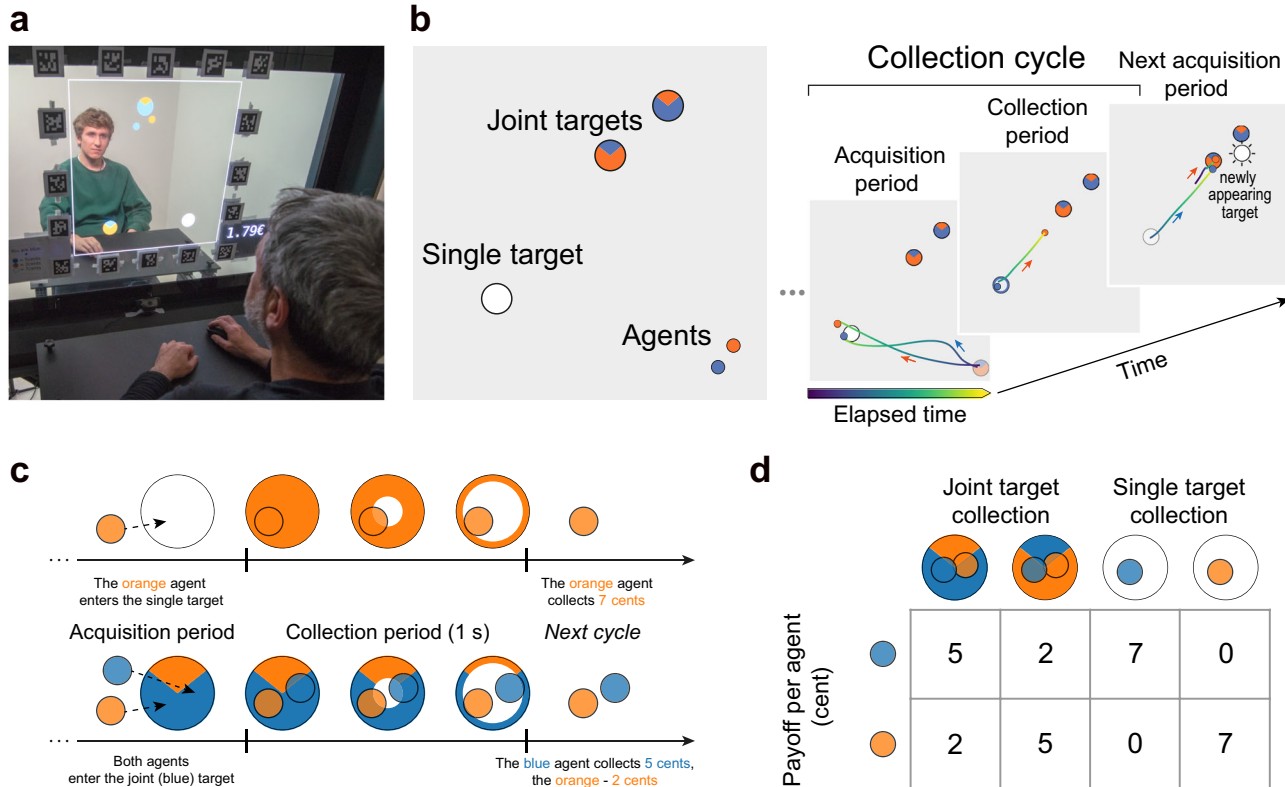

**Fig. 1 | Experimental setup and the game.** See Supplementary Movie S1, YouTube, OSF. **a** Two participants playing the Cooperation–Competition Foraging (CCF) game on a transparent OLED screen, in front of each other. Note: the people depicted here are authors, the people shown in the Supplementary Movie S1 are lab members, and all have provided explicit consent and are only shown for illustrative purposes. **b** Left: Game depiction. Small blue and orange circles are the two cursors ("virtual agents") controlled by the participants with a computer mouse. Agents collect targets (larger circles) by hovering over them. Each agent can collect the white target ("single target") on their own, while the colored targets ("joint targets") can only be collected cooperatively—when both agents hover over it simultaneously. If both agents arrive at a single target, the agent who first reaches the target wins. Right: Game progression. Each collection cycle begins with an acquisition period that lasts until one or both agents select a

target. During the subsequent collection period, if a single target is collected as in this example, the free agent can move around. Immediately after the target's disappearance at the end of the collection, the target reappears at a random position, and the next cycle begins. The color of trajectories represents elapsed time from the start of the period (visualizing the relative timing of the two agents: e.g., in the third frame the blue agent begins moving after the orange agent). **c** An agent (or both agents) enter the target and hover over it for 1 s to collect it. Once the collection of the white single target starts, it changes to the color of the collecting agent. The expanding transparent circle from the target's center indicates the collection progress. At the end of each collection cycle, the sound is played and the display of total earnings in Euro is incremented. **d** Payoff matrix. The payoffs of the two participants in each cycle depend on the type of target collection.

this estimation of the dependence of the payoff on the FST, we could obtain the potential payoff increase if the participants had played more competitively.

### Statistical analyses

All statistical tests were two-sided, and the data met, at least approximately, the key assumptions of the tests used. For the nonparametric Mann-Whitney U tests and paired Wilcoxon signed-rank tests, the median (Mdn) followed by the interquartile range (IQR: [quartile 1, quartile 3]) for each group or condition is reported; the effect size $r_{rb}$ was measured by the rank biserial correlation, and 95% confidence intervals (CI) for the effect size are provided. For the binomial tests, the Clopper-Pearson exact method for the 95% confidence intervals of the proportion was used. Pearson's product moment correlation coefficient $r$ and its 95% CI were used to report correlations; here data distribution was assumed to be normal but this was not formally tested.

To calculate the statistical significance of the coefficients of the GLM in each dyad, we used the Wald test and estimated the required variance matrix by inverting the Hessian matrix at the maximum likelihood estimate[65]. The resulting p-values were adjusted for multiple comparisons ($n = 58$) to control the false discovery rate using the Benjamini-Hochberg procedure.

Statistical tests were calculated in R version 4.4.2 and in Python 3.10.

### Reporting summary

Further information on research design is available in the Nature Portfolio Reporting Summary linked to this article.

## Results

### A transparent continuous dyadic foraging game

To study interactions in a controlled setting, we recruited 62 pairs of human participants (dyads) who sat face-to-face across a large transparent bidirectional visual display and played the foraging game together on a two-dimensional (2D) field (Fig. 1a; we recommend viewing the setup and gameplay demo videos for a clearer understanding of the experimental procedures and interaction dynamics: Supplementary Movies). The game reflects the real-world nature in being continuous—both in time and in 2D space—and in enabling the dyads to choose between various levels of cooperation or competition when foraging. Hence, we name it a *Cooperation–Competition Foraging* (CCF) game. Each participant used a mouse-controlled cursor as a virtual agent visible to both participants. At any moment, there were three randomly positioned targets on the screen (Fig. 1b, left) visible to both participants: one "single target" that could be collected by a single agent (worth 7 cent), and two "joint targets" that could

**Fig. 2 | Each dyad converges to a specific stable strategy on the cooperation–competition spectrum. a** Moving average (1 min window) of the *fraction of single targets* (FST) for three representative dyads. (**b**) Mean absolute deviation from a dyad's eventual stable strategy as a function of time (shaded band represents 95% confidence interval). After 14 min, all 58 included dyads have converged to a stable FST. **c** Distribution of the stable FST across dyads, which we categorize into three groups: largely cooperative, preferentially collecting joint targets (FST ≤ 0.1, n = 14, Supplementary Movie S2, YouTube, OSF), largely competitive, preferentially collecting single targets (FST ≥ 0.9, n = 14, Supplementary Movie S4, YouTube, OSF), and an intermediate, performing mixed collections (0.1 < FST < 0.9, peaking around ≃ 1/3, n = 30, Supplementary Movie S3, YouTube, OSF). Colors along the vertical axis represent the stable FST of each dyad, from brown to cyan. Here and in (**d**), the non-circle markers (plus, diamonds) indicate special strategies described later. **d** First-minute FST vs stable FST (from 10−40 min). Most dyads decrease their FST (i.e. become more cooperative) over time.

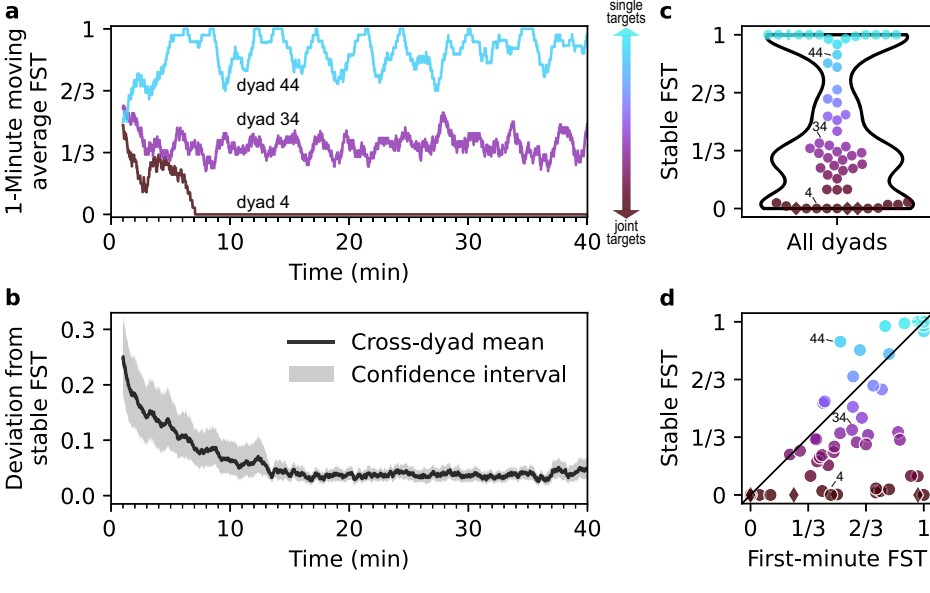

only be collected together, but have asymmetric payoffs (providing 5 cent to one, and 2 cent to the other agent, or vice versa) (Fig. 1c, d). By design, only one target can be collected at any time. Once a target is collected—requiring one agent for single targets or both agents for joint targets to remain on it for one second—it reappears at a random location on the playing field, initiating the next collection cycle (Fig. 1b, right). Crucially, the locations of the agents and the remaining two targets are not reset, preserving spatial continuity and allowing interactions to unfold naturally across successive cycles.

Although the single targets are designed to elicit a competitive element, we do not call them "competitive" because they can also be used cooperatively (see later). Nonetheless, since such cooperative strategy appeared only in one dyad, we refer to the dyads that mainly collected single targets as competitive, and to those that mainly collected joint targets as cooperative if not stated otherwise.

To avoid introducing an a priori bias towards the single or joint targets, we assigned the same *joint payoff* (i.e. the sum of payoffs for both participants) for both target types. Participants were informed about the corresponding payoffs (Fig. 1d), and were instructed to collect as many targets as possible, to earn their payment. To make the game more realistic by introducing a "travel cost", and avoid a total payoff being uniquely dependent on skill, the maximal movement speed was limited. If the limit was exceeded, the agent's position (cursor) lagged relative to the mouse until they slowed down.

After a short initial practice to familiarize themselves with the game mechanics, participants played two 20-min blocks and were rewarded by the cumulative payoffs collected in one randomly chosen block. By embedding foraging in a shared virtual space and a salient social context, our setup provides a controlled yet dynamic environment to study how dyads develop cooperative or competitive strategies.

## Dyads converge to stable strategies on the cooperation–competition spectrum

Due to the continuous and open nature of the game, we expected a variety of strategies to emerge. Indeed, most dyads, after an initial transient, converged to a specific set-point on the cooperation–competition spectrum, as represented by a relatively stable *fraction of single targets* (FST; the number of single targets collected over a certain period divided by the number of all targets collected over the same period; Fig. 2a). We found that all but 4 (58/62) dyads exhibited a stable FST after the 14 min period (Fig. 2b), and most dyads converged

within 10 min. Therefore, we excluded the first 10 min and used the remaining relatively stable 30 min of interaction (1.5 blocks) for our analysis focusing on stable strategies. The 4 dyads that abruptly changed their FST after the initial convergence period were excluded from further analysis. Interestingly, most dyads that changed their FST during the initial period became more cooperative during this time (Fig. 2d). After convergence, dyads were distributed along the entire FST axis (Fig. 2c). But the FST distribution is not uniform—dyads can be categorized into three "groups" along the continuum of FST: (1) cooperative dyads that mainly coordinate to collect joint targets (FST ≤ 0.1), (2) dyads that mostly compete for single targets (FST ≥ 0.9), and (3) dyads with intermediate, yet stable, strategies (0.1 < FST < 0.9, with a peak around FST ≃ 1/3). The spontaneous emergence of the three apparent groups raises the question about the strategies underlying the choices along the cooperation–competition spectrum. In what follows, we show that cooperative dyads use across-cycle history effects and leader-follower dynamics to coordinate on joint targets; competitive dyads race to single targets and often employ strategic positioning; and intermediate dyads frequently select the closest target, but also draw on interaction history and sensorimotor invitations to cooperate.

## Path minimization and cooperation–competition ratio shape dyadic strategies

To build a theoretical foundation for describing the observed strategies, we derive the optimal dyad strategies (in terms of joint payoffs) under different assumptions. According to the optimal foraging theory[53,57], and economic decision theories[51,66–70], foraging agents should maximize reward and minimize effort. We expand on these principles for the case of dyadic decisions.

For idealized agents that move in a straight line at the maximal possible speed (which is limited by game design), the distance to the collected target determines the payoffs that can be obtained within a fixed time. If one assumes that both agents always share the same position at the start of each collection cycle, then selecting the closest among three potential targets implements path minimization (Fig. 3a). Minimizing the path leads, due to the random target placement, to an optimal FST of exactly 1/3 (Fig. 3c, gray curve, middle purple square).

However, an identical positioning of both agents at the start of every collection cycle is not the best strategy for optimizing the joint payoff. Instead, during the collection of a single target, the non-collecting (free)

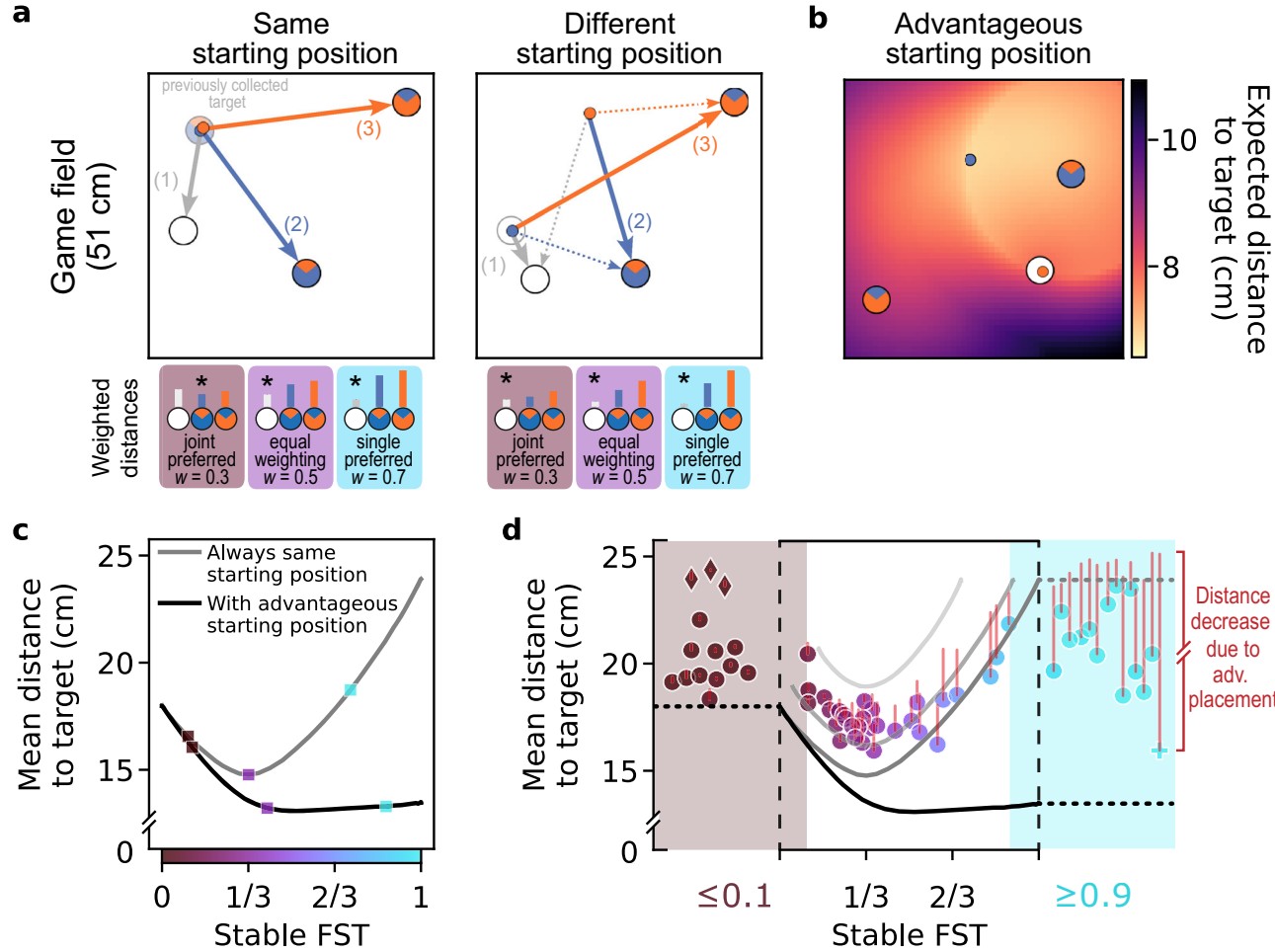

**Fig. 3 | Weighted path minimization and advantageous placement. a** Two example initial conditions. Three distances are relevant for target selection (solid arrows): for the single target, the shortest distance (1) from either agent; for each joint target, the longest distance (2, 3) from either agent. Irrelevant distances are shown as dotted arrows. Examples of "weighting" these distances by different FST preferences are shown below, where the target with the shortest weighted distance (indicated by asterisks) is selected. **b** Example of advantageous placement for one spatial configuration. While one agent collects a single target, the free (non-collecting) agent (blue in this example) strategically moves into a starting position that minimizes the expected distance to the next target, as indicated by the colormap (see Supplementary Methods 1.1 and Supplementary Fig. S1 for details). **c** Simulated optimal strategies. Varying the weighting of distance-based preferences produces different FST levels and mean distances to collected target. The markers correspond to the weights illustrated in (**a**). Simulations assume either the same starting position as in the left panel in (**a**) (gray curve) or advantageous placement (black curve). **d** Mean distance to collected target in simulated strategies (curves) and actual dyads

($n = 58$, markers). In addition to the simulations from (**c**), two lighter gray curves represent simulations with the same starting position but when the weighted closest target is chosen in only 70% (upper gray curve) or 90% (middle gray curve) of collection cycles, and a random selection otherwise. The markers represent the observed mean distance to the target for each dyad; the red vertical lines indicate the mean distance reduction due to the specific degree of advantageous placement performed by each dyad. Note that the actual dyads' data contain, in contrast to the simulations, a jitter due to the limited amount of target collections. Intermediate dyads span the space between 90% and 70% simulated strategies when the distance reduction due to advantageous placement is subtracted (top of red lines). Mostly cooperative dyads (FST ≤ 0.1, brown markers and shading) and mostly competitive dyads (FST ≥ 0.9, cyan markers and shading) are plotted separately to illustrate larger deviations from the weighted path minimization, such as strict turn-taking between the two joint targets (diamond markers) and varying use of advantageous placement.

agent can place itself to minimize the expected distance to the next target —thereby contributing cooperatively to joint efficiency, *across* collection cycles. Such "advantageous" placement must satisfy two conditions. First, it should minimize the expected distance from *either* agent to the next randomly appearing single target (Fig. 3b and Supplementary Fig. S1a). Second, the free agent should not place itself further from the nearest joint target than the currently collecting agent (e.g., stay within the circle around the "blue" joint target in the example in Fig. 3b). Thereby, the free agent does not delay a potential subsequent collection of the joint target (Supplementary Methods 1.1). The combination of both conditions results in a non-uniform landscape with a minimum, such as the map shown in Fig. 3b. As a consequence of such advantageous placement, the two agents together cover a larger area any agent can reach within a limited time, increasing the probability that collecting the newly

appearing single target will be a better choice than collecting a joint target. Therefore, higher FST values are now yielding better results, with an optimum at FST ≈ 0.55 instead of 1/3 (simulation results in Fig. 3c, black curve, see also Supplementary Fig. S1 for details).

It is important to emphasize that this advantageous placement strategy is *not* competitive. It maximizes the joint payoff of a dyad. It is thus a cooperative placement minimizing the expected distance to the next target; but during the acquisition period, both agents might compete again. In a fully competitive strategy, the free agent would not aim at minimizing the expected distance from *either* agent to the single target. Instead, the free agent would optimize the probability to be nearer to the newly appearing single target than the collecting agent. In practice, this leads to a placement near to the collecting agent, but a bit closer to the center of the game field (Supplementary Fig. S1f).

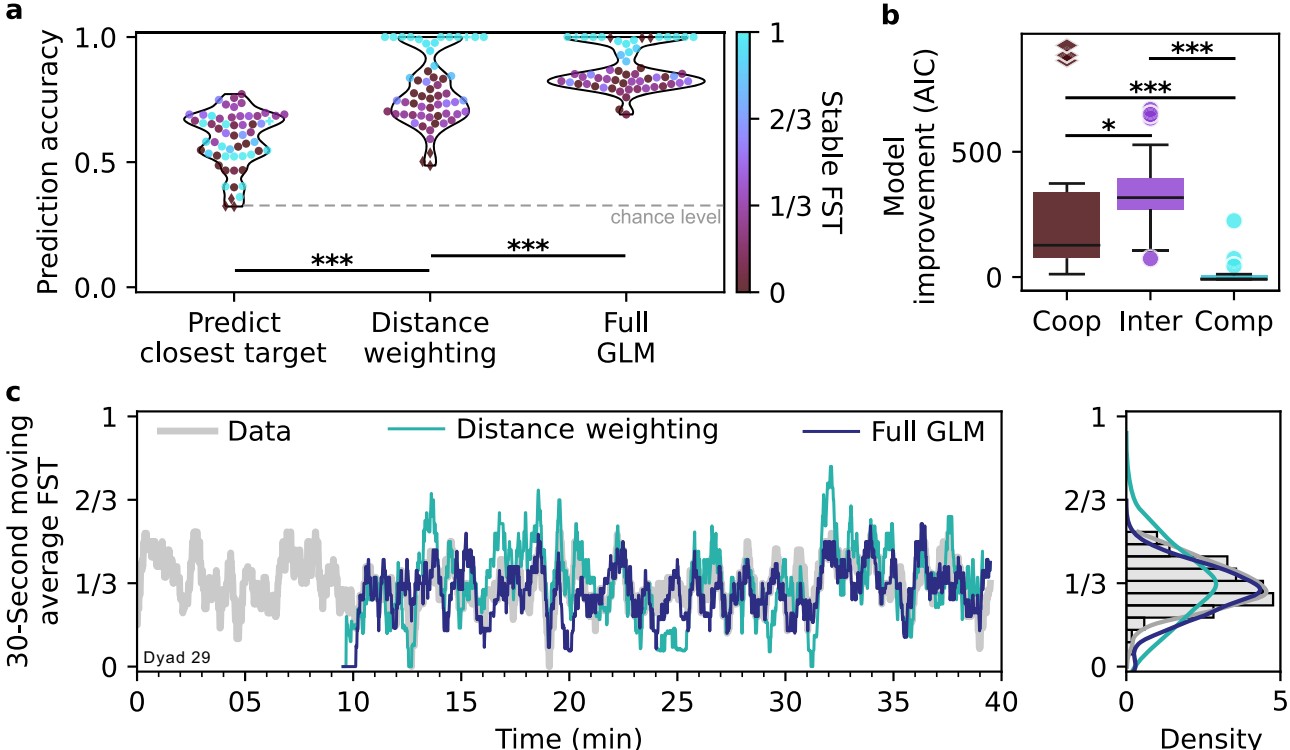

**Fig. 4 | Cooperation/competition-weighted path minimization and across-cycles predictors explain dyadic target choice. a** Predicting the choice of the next target, using (i) the closest distance, (ii) the weighted closest distance or (iii) the "full" generalized linear model (GLM). The full GLM, which includes across-cycles predictors such as target choice history and *invitations*, explains the target choices across all dyads ($n = 58$) in 87% of collection cycles. **b** Model improvement (Akaike information criterion, AIC) when including across-cycles predictors, shown as box-and-whisker plots (median, interquartile range, min/max, and outliers). Integrating recent choice and social information, represented by across-cycle predictors, improves the prediction in the cooperative group (Coop, $n = 14$) and is even more pronounced in the intermediate group (Inter, $n = 30$), compared to the competitive group (Comp, $n = 14$). **c** Timecourse of actual (gray curve) and predicted FST (moving average over target choice prediction, teal and blue curves) in one exemplary dyad. While the random target placement drives the fluctuations around the mean FST (left panel), incorporating the full GLM better matches the observed variance (right panel, see Supplementary Fig. S3 for population data).

Despite these clear theoretical optima, actual dyads spanned the entire range of FSTs (Fig. 3d), thereby deviating from a simple path-minimizing strategy. Therefore, we explore optimality under the constraint of a specific FST. For each dyad, we introduced a specific weighting factor, representing their manifested preference to choose single versus joint targets (Supplementary Fig. S2, "target type weighting"). This weighting factor reflects the overall preference of a dyad for joint or single targets, typically approximating their preference for cooperation or competition. For example, if a dyad prefers joint targets over single targets, they might choose a joint target even when the single target was physically the closest (Fig. 3a, left panel, brown rectangle). If however the single target is very close, then even a dyad that strongly prefers joint targets might occasionally select the single target (Fig. 3a, right panel, brown rectangle).

For each FST, we calculated the optimal strategy—the minimal distance attainable (Fig. 3c), either by assuming simple path minimization (gray curve) or additionally taking into account advantageous placement of the free agent (black curve). While for simple path minimization there is a clear optimum at FST 1/3, there exists a continuum of similarly good strategies for path minimization with additional advantageous placement (0.4≤ FST ≤1; note the nearly flat black curve starting at 0.4 in Fig. 3c). Most actual dyads, however, performed advantageous placement only to a certain degree (Fig. 3d and Supplementary Fig. S1g). Therefore, the equal weighting path minimization without advantageous placement can explain the formation of the intermediate group around FST 1/3. For each dyad, we computed the mean distance to the selected target across trials; statistical comparisons between groups were then performed on the distribution of these per-

dyad means using nonparametric tests, which assess differences in group-level medians. Indeed, the distance to target in the intermediate group is reduced compared to the cooperative and the competitive groups (Mann-Whitney $U$ test, intermediate vs cooperative: $U = 388$, $p < 10^{-5}$, $n_1 = 30$, $n_2 = 14$, $Mdn_1 = 17.52$, $IQR_1 = [17.00, 18.30]$, $Mdn_2 = 20.22$, $IQR_2 = [19.36, 21.72]$, $r_{rb} = 0.68$, $CI = [0.51, 0.8]$; intermediate vs competitive: $U = 367$, $p < 10^{-4}$, $n_1 = 30$, $n_2 = 14$, $Mdn_1 = 17.52$, $IQR_1 = [17.00, 18.30]$, $Mdn_2 = 20.78$, $IQR_2 = [19.63, 22.21]$, $r_{rb} = 0.60$, $CI = [0.31, 0.79]$; all tests in this paper are two-sided).

To estimate how well the dyads follow a (weighted) path minimization strategy, for each collection cycle we predicted the subsequent choice of the target (Supplementary Methods 1.1). For non-weighted path minimization, the average choice prediction accuracy across dyads was 59%, and the median 61%, significantly higher than the 33% chance (Fig. 4a, Wilcoxon signed-rank test, $W = 3$, $p < 10^{-6}$, $n = 58$, $Mdn_1 = 0.61$, $IQR_1 = [0.53, 0.68]$, $Mdn_2 = 0.33$, $IQR_2 = [0.33, 0.33]$, $r_{rb} = 0.87$, $CI = [0.85, 0.87]$). The dyads with intermediate strategies were particularly well predicted (average accuracy 65%; Mann-Whitney $U$ test comparing accuracies of intermediate and non-intermediate dyads, $U = 740$, $p < 10^{6}$, $n_1 = 30$, $n_2 = 28$, $Mdn_1 = 0.68$, $IQR_1 = [0.62, 0.69]$, $Mdn_2 = 0.53$, $IQR_2 = [0.45, 0.58]$, $r_{rb} = 0.65$, $CI = [0.49, 0.79]$), suggesting that the intermediate dyads often perform true (non-weighted) path minimization. For the weighted path minimization, the average choice accuracy increased to 78% across all dyads (Fig. 4a, Wilcoxon signed-rank test comparing non-weighted vs weighted minimization, $W = 21$, $p < 10^{-6}$, $n = 58$, $Mdn_1 = 0.61$, $IQR_1 = [0.53, 0.68]$, $Mdn_2 = 0.75$, $IQR_2 = [0.69, 0.88]$, $r_{rb} = 0.85$, $CI = [0.79, 0.87]$). Thus, a substantial portion of the observed dyadic strategies can be accounted for by combining simple

path minimization with differential weighting of joint versus single targets.

## Beyond weighted path minimization: social and sensorimotor planning factors

Although weighted path minimization—both within and across collection cycles—already accounts for 78% of choices, we identified additional factors that further enhance predictive accuracy. Analyses revealed that actual moving average FST fluctuations are typically smaller than expected from the weighted path minimization (Fig. 4c, Supplementary Fig. S3a), suggesting that dyads use more sophisticated strategies. Specifically, during single target collections, the free agent that is currently not collecting the single target often positioned itself onto one of the joint targets, effectively *inviting* the subsequent, cooperative joint target collection in the next cycle (Supplementary Movie S5, YouTube, OSF). Another prevalent pattern exhibited by many dyads in the cooperative and intermediate groups was a tendency to avoid the newly appearing target.

To account for these patterns, we incorporated invitations and the identity of the two previously collected targets as across-cycle predictors in a generalized linear model (GLM), in addition to the weighted path minimization. This model, calculated separately for each dyad, predicts the next collected target at the start of each acquisition period. The mean prediction accuracy improved compared to the weighted path minimization from 78% to 87% (Fig. 4a, Wilcoxon signed-rank test, $W = 14$, $p < 10^{-6}$, $n = 58$, $\mathrm{Mdn}_1 = 0.75$, $\mathrm{IQR}_1 = [0.69, 0.88]$, $\mathrm{Mdn}_2 = 0.84$, $\mathrm{IQR}_2 = [0.81, 0.98]$, $r_{rb} = 0.79$, $CI = [0.70, 0.84]$). The model improvement (AIC) due to inclusion of across-cycle predictors was apparent in the cooperative group in contrast to the competitive group (Fig. 4b, Mann-Whitney $U$ test, $U = 15$, $p < 10^{-4}$, $n_1 = 14$, $n_2 = 14$, $\mathrm{Mdn}_1 = 127$, $\mathrm{IQR}_1 = [75, 338]$, $\mathrm{Mdn}_2 = -8$, $\mathrm{IQR}_2 = [-8, 7]$, $r_{rb} = 0.72$, $CI = [0.49, 0.84]$), and it was even more pronounced in the intermediate group ($U = 128$, $p < 0.05$, $n_1 = 14$, $n_2 = 30$, $\mathrm{Mdn}_1 = 127$, $\mathrm{IQR}_1 = [75, 338]$, $\mathrm{Mdn}_2 = 317$, $\mathrm{IQR}_2 = [271, 395]$, $r_{rb} = 0.31$, $CI = [0.03, 0.64]$).

The GLM coefficients quantifying the effect of invitations are statistically significant in 30 dyads, with 29 showing a positive effect (Wald tests, here and further: $p < 0.05$, Benjamini-Hochberg-adjusted across 58 dyads). Thus, there is a significant increase in the collection probability of the invited joint target across dyads (binomial test, *proportion* = 0.97, 95% $CI = [0.82, 0.99]$, $Z = 4.9$, $p < 10^{-6}$). Note that in the case of an invite, the previously collected target is a single target that is subsequently avoided. The effect of avoiding the previous target is also clear if the previous target is a joint target (Wald tests, $p < 0.05$ in 37 dyads) but is inconsistent when the previous target is a single target and no invite is present. In the latter case, the GLM coefficients indicate a significant (Wald tests, $p < 0.05$) tendency to avoid the previous single target for 14 dyads but also a significant increase in the single target collection probability for 9 dyads (no consistent effect across dyads, binomial test, *proportion* = 0.61, $CI = [0.38, 0.80]$, $Z = 0.83$, $p = 0.43$). Thus, beyond cooperation/competition-weighted path minimization, additional planning factors across cycles—invitations and previous target identities —shape the dyadic strategies.

The prediction improvement introduced by these additional factors is also apparent in the time course of the actual and predicted FST. In contrast to the weighted distance prediction, the "full" GLM captures the dyads moving average FST fluctuations better (Fig. 4c, Supplementary Fig. S3, Wilcoxon signed-rank test comparing the differences of standard deviations between 30-second moving average FST of the actual and each of the two model's predictions for the 40/58 dyads that exhibited FST fluctuations, $W = 258$, $p < 0.05$, $n = 40$, $\mathrm{Mdn}_1 = -0.01$, $\mathrm{IQR}_1 = [-0.03, 0.02]$, $\mathrm{Mdn}_2 = 0.0008$, $\mathrm{IQR}_2 = [-0.009, 0.02]$, $r_{rb} = 0.32$, $CI = [0.04, 0.62]$). In other words, with few exceptions dyads primarily follow cooperation–competition weighted path minimization, but if the random target placement sequence happens to dictate a substantial deviation from their established mean FST, the across-cycle predictors come into play. For example, if weighted path minimization prompts several single target

collections in a row, participants would perform an invite leading to the collection of a joint target, and thus avoid a potential breakdown of established cooperation.

These findings highlight the influence of social and sensorimotor planning factors that extend beyond a cooperation–competition weighted path minimization. We propose that the observed reluctance to repeat the same target type reflects a dual contribution: a sensorimotor bias favoring prospective planning and coordination for already visible targets[47,71], and a social motivation for fairness. The latter is especially pronounced in the extreme case of the three exclusively cooperative dyads (FST = 0), who exhibited strict normative turn-taking between the two joint targets regardless of distance (Fig. 4a, diamond markers, Supplementary Movie S6, YouTube, OSF).

Likewise, the invitations represent one of the most basic forms of *sensorimotor communication*[72]. During the collection of a single target— primarily a competitive act—the non-collecting agent conveys a compelling social signal for cooperation by placing itself on a joint target. Indeed, our findings reveal that these invitations are accepted in the majority of cases (88%), even when the new single target is closer (79%). This indicates that beyond path minimization, invitations play an important additional role in shaping intermediate strategies, highlighting the interplay between social signaling and strategic decision-making.

## Choice certainty shapes ensuing spatiotemporal interactions

Thus far, our analyzes dealt with the prediction of discrete target choices at the beginning of each collection cycle. However, these choices are the result of sensorimotor interactions in the continuous action space. Here, we link the choice modeling to the classification of spatiotemporal trajectories, to characterize ongoing decision processes and dyadic coordination. Beyond the prediction of the discrete target choices in each cycle, we derive the certainty of this prediction, measurable via its entropy. This choice certainty represents an estimate of how sure the participants are about their next move, given the current spatial contingencies and dynamics across collection cycles as modeled by the GLM. Note that for this analysis, we exclude the dyads that mostly collected single targets (FST ≥ 0.9) since there is always a high certainty in their target choice. To relate target choice certainty to ensuing dynamics that precede and determine the subsequent collection, we analyzed the ongoing decision processes reflected in spatiotemporal trajectories. We classified the collection cycles into several representative classes, associated with different levels of apparent coordination (Fig. 5a; Supplementary Methods 1.3).

One prominent class of high coordination is the already mentioned *invitations*, where one agent places itself on a joint target in advance. This invitation could be reciprocated—the other agent moving towards it—or denied, leading to a "failed invitation". By definition, the invitations can only take place in dyads that collect single targets as well as joint targets from time to time. The next two classes are characterized by a *movement towards the same* (either single or joint) target. Following the collection of the previous target, the agents move simultaneously to the next target ("concurrent to the same target"), or one agent leads and the other follows ("one ahead to the same target"). Third, the trajectories could be "strongly curved", due to starting a preemptive movement before the choice is made[43,44], initial miscoordination, or multiple changes of mind (Supplementary Movie S7, YouTube, OSF). Lastly, one of the agents could select a joint target while the other would go to collect a single target (denoted here by "different targets").

The above classification is performed separately for each collection cycle. Given the continuous transition between the cycles, we explored the across-cycles transition probabilities, using Markov chain representation. The most prominent node is the "concurrent to same target": it is the most probable class after any type of interaction except when the invitation is present but not yet reciprocated. These miscoordinations, "different targets" and "failed invitation", elicit a social pressure and are typically corrected by the subsequently accepted "invitation". Such invitations are often *passive* (54% of all invitations)—the inviting agent communicates by remaining on the joint target they entered previously. The other 46% are *active* invitations,

**Fig. 5 | High uncertainty of target choice increases the probability of uncoordinated trajectories.**
**a** We classify the trajectories within each collection cycle into different classes (outer panels; the color of the trajectory represents the time from the start of the collection cycle, the faded targets are the targets collected in the preceding cycle). Transition probabilities are shown using a Markov chain (inner panel, the intensity of the arrows indicates transition frequencies). The color of the nodes (circles) corresponds to one of the six classes. **b** Distinct fractions of trajectory classes are observed across varying levels of target choice uncertainty, as estimated by the full GLM. Trajectories associated with miscoordination or failed prediction of the partner's choice ("Strongly curved" and "Different targets") are often apparent when the model has a higher uncertainty. In contrast, trajectories with high coordination between agents (accepted "Invitations" and "Concurrent to the same target") are predominant at low model uncertainties. Trajectory classes are denoted by the same color as in (**a**). **c** After an invite by sensorimotor communication the uncertainty about target choice is significantly reduced. **d** Average frequency of trajectory classes as a function of stable fraction of single targets (FST). Note that in (**a**) (inner panel), (**b**, **c**) only dyads with non-negligible target choice uncertainty are included (FST < 0.9, $n = 44$); in (**d**) all dyads are included ($n = 58$).

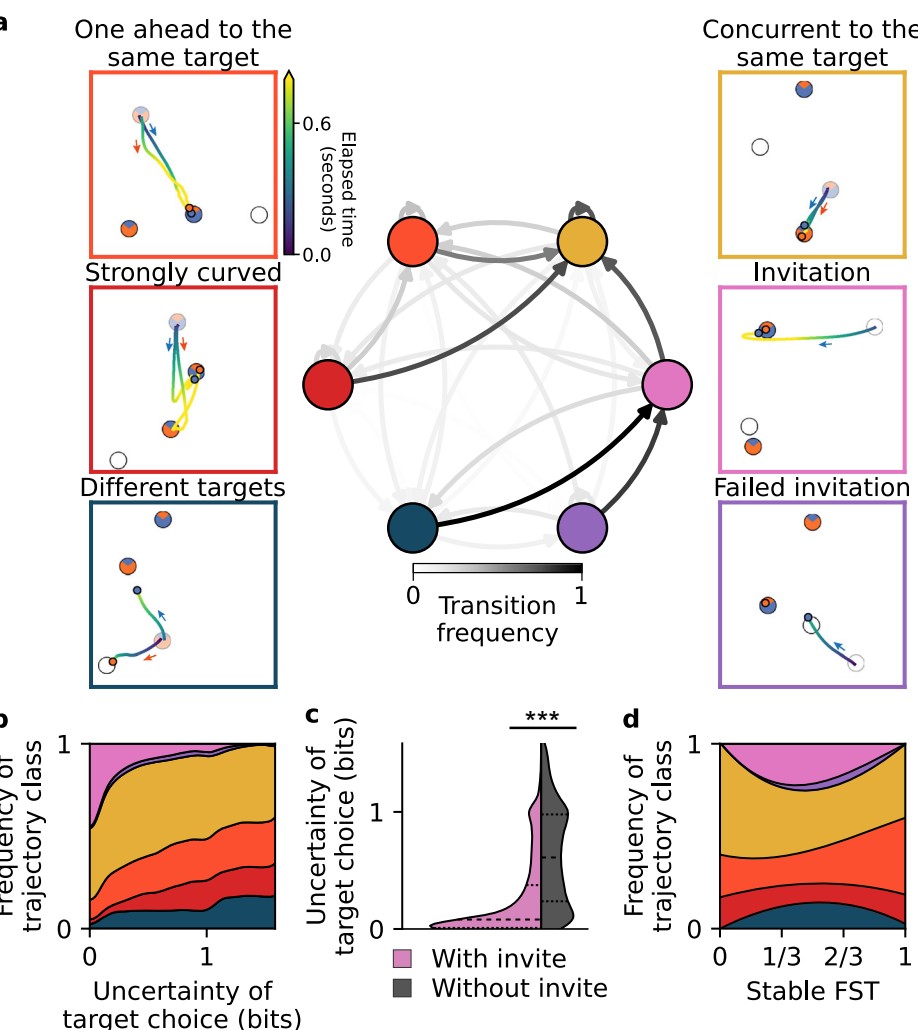

taking place after both agents aimed for the single target (concurrent or one ahead). Thus, we can divide the invitations into passive and active sensorimotor communication.

By relating the trial classes to the target choice uncertainty of GLM predictions, we found that higher uncertainty is associated with a higher prevalence of less coordinated interactions (Fig. 5b). In some situations of high uncertainty, dyads manage to maintain coordination, either by one agent moving ahead and signaling the next target ("one ahead to same target"), or by multiple trajectory adjustments leading to "strongly curved" trajectories. In other situations, however, high uncertainty leads to a breakdown of coordination ("different targets"). In contrast to these classes, the mean uncertainty of collection cycles classified as invitation is significantly reduced (Fig. 5c, Mann-Whitney U test comparing prediction entropy of initial conditions with and without the invite factor, Wilcoxon signed-rank test comparing prediction entropy of initial conditions with and without the invite factor, $W = 2297395$, $p < 10^{-6}$, $n = 5948$, $Mdn_1 = 0.08$, $IQR_1 = [0.01, 0.37]$, $Mdn_2 = 0.61$, $IQR_2 = [0.24, 0.98]$, $r_{rb} = 0.59$, $CI = [0.57, 0.61]$), demonstrating the utility of salient sensorimotor communication for efficient coordination.

The relation between spatiotemporal dynamics and uncertainty is also apparent in the observed distribution of collection classes over stable FST (Fig. 5d). One of the two classes associated with high uncertainty, "different targets", is peaking at intermediate FSTs, in line with high certainty of target type choice for low and high FST dyads. The other class, "strongly curved", is uniformly distributed because it encompasses multiple components: the uncertainty about which of the two joint

targets to select (low FST dyads), which of the three targets to select (intermediate FST dyads), and the uncertainty about position of the next single target (high FST dyads; "go-before-you-know" or "decide-while-acting" effect that refers to when the action starts before the goal is determined[44,47]; Supplementary Fig. S4).

More generally, Fig. 5d shows that in high FST dyads, both agents nearly always moved toward the same (single) target, indicating that these participants were indeed *competing*, rather than allowing one another to collect the target.

These analyzes show that our generalized linear model captures, to a large extent, the cognitive processing underlying the ensuing dyadic choices. At the same time, it is remarkable that even at a high model uncertainty, the agents still often exhibit straight, coordinated concurrent movements to the same target. This can be explained, at least in part, by co-evolving online coordination, whereby participants closely observe each other's movements and continuously adjust their trajectories.

**Spatiotemporal variables shape joint payoff**
As demonstrated in the preceding sections, real dyads do not always choose the (target type-weighted) nearest target, and they might perform advantageous placement only to a certain degree. Furthermore, real dyads do not always move in straight lines at constant speed. Here, we disentangle the contribution of these deviations from idealized patterns to the obtained payoffs.

In a first approximation, the main contributing factors to the joint payoff are the dyads' average trajectory length and their average speed,

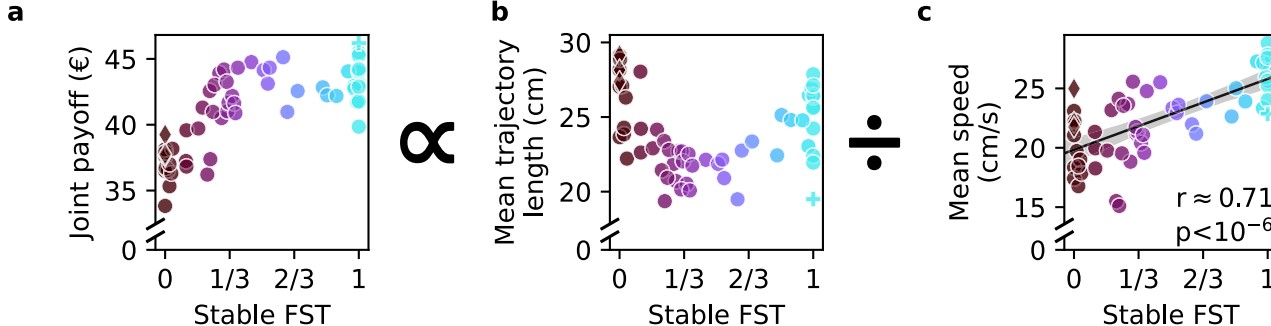

**Fig. 6 | Spatiotemporal factors shape the payoff in a continuous action space. a** The joint payoff across both participants in a dyad is proportional to the mean acquisition duration. The mean acquisition duration is well-approximated by dividing the mean trajectory length (**b**) by the mean movement speed (**c**) on these trajectories. Note the increase in speed with higher fractions of single targets (FST). Each marker represents one dyad (n = 58). See Supplementary Fig. S5 for the analysis that decomposes the mean trajectory length in (**b**) into three contributing components.

which explains why the joint payoff increases as a function of FST up to 1/3 and then stabilizes (Fig. 6a). The payoff in this task is proportional to the total amount of targets collected, which is again approximately proportional to the inverse of the mean collection cycle duration (Pearson's correlation coefficient $r(56) = 0.99$, $p < 10^{-6}$, $CI = [0.99, 0.99]$). The mean acquisition duration is well approximated by the limiting agent's mean trajectory length and speed ($r(56) = 0.99$, $p < 10^{-6}$, $CI = [0.99, 0.99]$). For single targets, the limiting parameters are the trajectory length and the speed of the agent who collects the target. For joint targets, these parameters are determined by the agent that enters the target last. The resulting trajectory length is a U-shaped function of FST (Fig. 6b). The speed increases with FST (Fig. 6c, $r(56) = 0.71$, $p < 10^{-6}$, $CI = [0.55, 0.82]$), demonstrating that more competitive dyads move faster (this result also holds for the average speed of the two agents rather than the agent that won the single target $r(56) = 0.59$, $p < 10^{-5}$, $CI = [0.39, 0.73]$). Due to the increased speed, the effect of longer trajectories at high FSTs on payoffs is compensated; therefore, this U-shaped function does not translate to the joint dyad payoffs—the payoff of high FST dyads does not drop (Fig. 6a).

To understand the U-shaped pattern in trajectory length, we decomposed it into three components: distance to the next target, curvature, and advantageous placement (Supplementary Fig. S5d-f). Most dyads followed path minimization, with distances falling between simulated scenarios with 70% and 90% closest (weighted) target. Curved trajectories were more common in dyads focusing on joint targets, often due to initial mis-coordination. Notably, turn-taking dyads (diamond markers) avoided miscoordination through consistent alternation. On the other extreme (FST = 1), some fast-acting dyads exhibited curved paths due to "go-before-you-know" behavior (Supplementary Fig. S4,[43]). Finally, the contribution of advantageous placement that shortens the anticipated path scaled up with FST but dyads only partially exploited the theoretically possible distance reduction. Some agents stayed close to their collecting partner (Supplementary Movie S4, YouTube, OSF), others positioned themselves strategically close to the center to compete (Supplementary Fig. S1f, Supplementary Movie S8, YouTube, OSF). One unique dyad (plus marker) efficiently split the field to collect single targets cooperatively (Supplementary Movie S9, YouTube, OSF).

In conclusion, this decomposition demonstrates how sensorimotor variables shape the joint payoff, leading to the lower payoff of cooperative dyads (Fig. 6a). Most cooperative strategies (FST < 1/3) come at the cost of (i) longer distances to the next target, (ii) lower speed, (iii) more curved trajectories for some dyads because of initial miscoordination, and (iv) the loss of the optimization opportunity by advantageous placement before the beginning of a collection cycle. Special cooperative cases include the highest scoring dyad that effectively split the field to share single targets, and strict turn-takers that move faster that most other dyads with equally low FST. Similarly, more competitive dyads

compensate for the longer distances by higher speed and advantageous placement optimizations.

## Payoff differences between participants, skill and cost of cooperation

Thus far, we considered the joint payoff and the observed strategies at the level of a dyad. However, the two participants in a dyad might differ in several regards, for instance in their ability—or willingness—to collect single targets, or due to a biased collection of joint targets benefiting one participant. Therefore, here we explore the factors underlying *within-dyad* payoff differences and assess the cost of deviating from individually optimal strategies.

Across dyads, within-dyad payoff difference (ranging from 0 to 14 Euro; Fig. 7a) correlated with the FST ($r(56) = 0.67$, $p < 10^{-6}$, $CI = [0.50, 0.79]$). This correlation was nearly perfectly accounted for by the difference in the number of single target collections between the two agents (Fig. 7b, $r(56) = 0.99$, $p < 10^{-6}$, $CI = [0.98, 0.99]$).

Assuming that if both agents move straight to the single target it reflects competition (see the classification in Fig. 5 and Methods), we identify the difference in single target collections that can be explained by the discrepancies in the *competitive* skill between the two members of a dyad. Such discrepancies accounted for the large part of single target collection differences (82%), especially in more competitive dyads (FST ≥ 0.9, 96%). The remaining part of the difference in single targets might reflect varying attitudes toward deviating from cooperation on joint targets in favor of opportunistically pursuing a nearby single target ("different targets" class, see Fig. 5). Those selections are not considered to be related to the skill difference and are not incorporated into subsequent analyses.

We numerically estimated the resulting individual payoff as a function of competitive skill difference and FST (Fig. 7c), assuming the average dyad's speed and trajectory length at each FST. For equal skill, the expected individual payoff corresponds to half of the joint payoff (Fig. 6a), showing a flat plateau between FST 1/3 and 1. A competitive skill difference leads to considerable payoff discrepancies at high FST. The estimated payoff at (-30%; 30%) skill difference qualitatively corresponds to the observed payoff difference range (Fig. 7a). Even small skill differences (above 6%) make a competitive FST = 1 strategy optimal for the more skilled participant. For the less skilled participant, the optimal FST is around 1/3 (see Supplementary Fig. S6b). The greater the skill difference, the more the individual participants' optimal strategies deviate from each other (Fig. 7c).

This analysis begs the question of how far the chosen strategy of each participant, i.e. their level of cooperation vs competition, deviates from the optimal given their skill. In particular, higher-skilled participants who chose cooperation could have increased their payoffs by adopting a competitive approach. Note that it only makes sense to consider the options of the

**Fig. 7 | Individual payoff, skill difference and cost of cooperation.** The optimal individual strategy that maximizes the individual payoff is determined by skill differences, and in general favors collecting single targets. Nevertheless, many participants chose a more cooperative strategy with joint targets and paid a cost of cooperation. **a** Individual payoff and difference between participants. Each bar represents the lower and the higher individual payoff in a dyad (bottom and top ends of the bar, respectively); hence, the bar length represents the payoff difference between the participants. If the payoff difference is below 30 cents a marker instead of a bar is used. Dyads ($n = 58$) are arranged in order of increasing FST. **b** The payoff difference within a dyad ($n = 58$) is mainly determined by the difference in single targets collected by each agent, and only little by a bias towards one or another joint target. **c** Estimated payoff depends on FST and skill differences within a dyad. The greater the competitive skill difference (i.e., the normalized difference in single targets collected in cycles which we assume to be competitive, see Methods), the more beneficial competitive strategies become for the higher-skilled participant. Note that the black curve (skill diff. = 0%) corresponds to the average profile for the joint payoff data shown in Fig. 6a, divided by two. **d** Estimated loss of payoff, representing the monetary "cost of cooperation" for the higher-skilled participant within a dyad ($n = 44$). Despite this cost, many participants still chose to cooperate (lower FST) rather than maximize their economic gain.

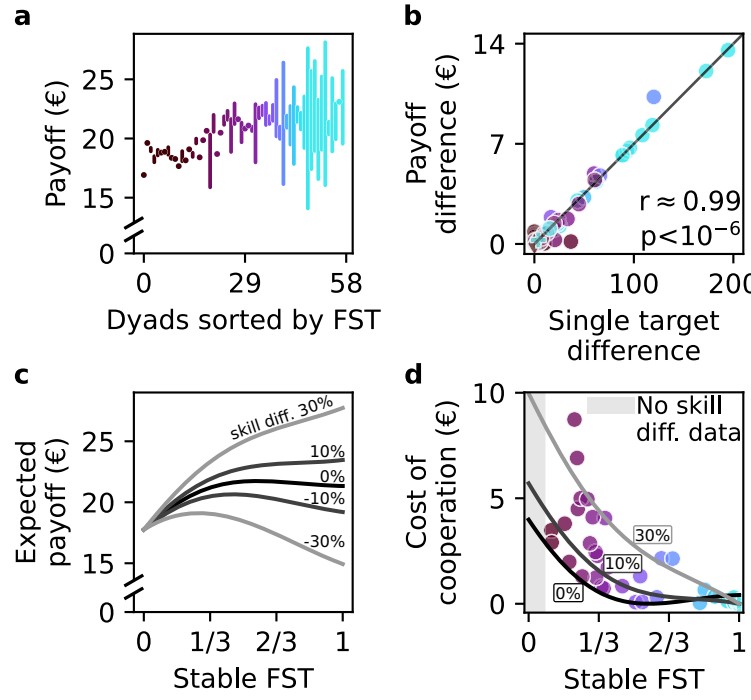

higher-skilled participants because only they can make a profitable unilateral decision to pursue single targets competitively instead of cooperation. To quantify the loss of payoff (the cost of cooperation) due to a suboptimal individual strategy, we use the estimated competitive skill difference within the dyad, assuming its independence of FST. Under this assumption, we can extrapolate the expected payoff to other FST values. For each higher-skilled participant, we calculate the loss of payoff as compared to the optimal FST = 1 scenario (Fig. 7d). The largest cost of cooperation we could estimate is as high as 8.7 Euro (>42% of individual earnings).

Overall, these results demonstrate that many participants do not follow their individually optimal strategies. All participants in the highly cooperative dyads at FST = 0 are suboptimal independently of their potential skill differences. That includes the lower-skilled participants, who would also benefit from increasing the amount of single target to around FST = 1/3 (Supplementary Fig. S6b). The intermediate dyads around FST = 1/3 are around the estimated optimum for the lower-skilled participant. Here the higher-skilled participants could simply increase the FST to exploit their optimal competitive strategy. Instead, however, they choose a more prosocial cooperative strategy. Interestingly, we did not observe a correlation between the skill difference within the dyad, and whether the dyad chose to follow a competitive or cooperative strategy (Supplementary Fig. S6a). This suggests that factors beyond the selfish monetary gains of the higher-skilled participant influence the choice of the final strategy.

In summary, our findings demonstrate the interplay between skill differences within dyads and the optimal trade-off between cooperation and competition. As skill disparities increase, the individually optimal strategies of participants diverge, with higher-skilled participants benefiting from competitive strategies and lower-skilled participants profiting more from intermediate strategies. Despite these differences, many dyads chose to incur a cost of cooperation, favoring more prosocial strategies over maximizing their economic gain. This deviation from the optimal strategies underscores the complexity of decision-making in social

settings, where factors beyond individual reward and effort, such as prosocial preferences or adherence to previously established cooperation, influence behavior.

## Discussion

Using a continuous transparent foraging game and behavioral modeling, we found that human dyads converge on distinct strategies along a cooperation–competition spectrum. Some dyads competed for winner-take-all single targets, others cooperatively shared joint targets, and many adopted intermediate strategies. A model incorporating dyad-specific target weighting, movement efficiency, and interaction history captured much of this variability, with intermediate dyads often relying on sensorimotor cues such as invitations to coordinate. The certainty of model predictions was linked to spatiotemporal dynamics of mutually coupled sensorimotor decisions. Beyond decision processes, we identified systematic sensorimotor influences on payoff distribution: competitive dyads benefited from faster movements, many intermediate dyads achieved comparably high payoffs through efficient path minimization, whereas largely cooperative dyads earned less. Notably, many higher-skilled participants nonetheless adopted more cooperative strategies than individually optimal, thereby accepting a cost of cooperation.

Historically, interaction experiments have been conducted using social dilemmas from game theory in a rigorous context but discrete in time and space. Compared to such discrete decisions, a tighter integration of continuous decisions with physical effectors such as eyes and limbs makes them more susceptible to sensorimotor affordances, as actions are not only planned but are iteratively recalibrated through direct sensory feedback[39,45,73]. Recent paradigms adopted more continuous approaches that resemble real-world decision-making[19,27,35,39]. Prior work already leveraged cooperative[23,28,74], as well as competitive interactions[31,32,75,76], into continuous action spaces. Similarly, in our foraging experiment participants could directly observe and react to their partner's ongoing decisions, embedding the interaction within embodied, real-time sensorimotor flow. The face-to-

face arrangement and spatial continuity between collection cycles provided an additional layer of naturalism.

However, this transition towards continuity happens not only on the level of action spaces but also on the type of social interaction. While typical experiments offer only a fixed social context—either a cooperative, a competitive, or at most a binary dichotomy between these two—recent work by Pisauro and colleagues has began to investigate the continuous trade-off between these extremes along a 1D axis[16]. In their Continuous Dilemma study, the shifts from cooperation to competition were elicited experimentally by the change of payoff formulation. Our work extends this emerging line of research by combining direct action visibility in a continuous 2D action space with the opportunity to manifest graded, spatially-dependent cooperative, intermediate, or competitive strategies, all within the same "neutral"—if actual spatiotemporal contingencies are disregarded—social context.

## Emergent strategies and underlying decision processes

Conceptually, our paradigm synthesizes ideas from classical games such as Stag Hunt (coordination required to collect joint targets), Battle of the Sexes / Bach or Stravinsky (asymmetry of coordination options), and Prisoner's Dilemmas (choice between cooperation and competition), transferring them onto a continuous spatiotemporal interaction in a two-dimensional action space. Using this foraging paradigm, we recorded over 40 h of continuous dyadic interactions that spanned the entire cooperation–competition spectrum. Strategies emerged not only at the extremes of pure cooperation and competition but also within a range of intermediate strategies. Our analysis indicates that travel path minimization underlies the emergence of this intermediate group. Generalizing this path minimization, by weighting target types based on stable cooperative or competitive tendencies in each dyad, we successfully characterized the large part of emergent strategies. Extending our model by recent interaction history and invitations (sensorimotor communication)[72,77–80] across collection cycles further enhanced prediction accuracy. Due to spatial continuity between cycles, a non-engaged participant can signal their intention to cooperate by inviting the partner to select a specific joint target. As hypothesized, the prediction improvement was more apparent in cooperative dyads compared to predominately competitive ones. However, contrary to our initial expectations, the effect was even stronger in the intermediate group. Thus particularly within the intermediate group, where decisions are made under constant tension between cooperative and competitive dynamics, the generation and integration of social information are increased. We speculate that invitations, especially the "active invitations" that take place after both participants tried to collect the single target, serve to relieve the social pressure after the competitive episode, and can be framed as a measure of trust in cooperative reciprocation by the partner.

Another consequence of continuity between cycles is a possibility for the free agent to position itself strategically during single target collection by the other. Depending on the spatial configuration and the chosen location, such advantageous placement may benefit either participant—and thus reflect a trust in eventual reciprocation—or gain a competitive edge for the free agent. In practice, most dyads who employed the advantageous placement showed a mix of these modes, likely reflecting limited understanding of optimality and reliance on heuristics to gain timing advantages. This ambiguity makes it difficult to infer motivation from spatial behavior alone, without incorporating movement skill and reaction times into a model.

Linking across-cycle dynamics to within-cycle effects, we demonstrated that target choice uncertainty significantly shapes ensuing spatiotemporal interactions, leading to either mutually coupled sensorimotor dynamics or miscoordination when social information is unavailable and uncertainty is high[39,75]. By quantifying the utility of sensorimotor communication as a reduction in target choice uncertainty, we found that intermediate dyads effectively leveraged this signal to arbitrate between cooperative and competitive strategies. The emergence and strategic use of sensorimotor communication for coordination are key outcomes of our continuous game design[79,81,82], distinguishing it from traditional discrete

paradigms. These findings directly address the open question of how, and in which interactive scenarios, the sensorimotor communication arises, demonstrating that continuous, adaptive interactions are crucial for its emergence[72].

Since each collection cycle ends with the selection of one of three targets, it might seem that, similar to discrete mixed-motive games, decisions in our paradigm are reduced to a binary—or trinary, if the type of joint target is considered—choice between cooperation and competition. From this perspective, the gradual ratio of cooperation and competition is achieved only across multiple interactions. However, we argue that, unlike discrete paradigms, decisions in our task are shaped by a varying context determined by distances to targets and the partner. For instance, choosing a distant joint target over a nearby single target reflects a higher degree of cooperation than selecting a very close joint target. Additionally, classification can sometimes be misleading—e.g., participants may attempt to converge on a joint target but fail to coordinate, ultimately leading to a single target collection. In such cases, despite the final competitive outcome, the underlying interaction was mostly cooperative. Thus, in our paradigm varying levels of cooperation and competition can manifest not only across multiple but also within each cycle.

Our dyadic foraging game differs from classical foraging paradigms, which typically focus on the trade-off between exploiting the current resource and exploring new options under constraints of depleting rewards and the cost of exploration[83–85]. In our task, the only limited resource is the finite duration of the session, meaning that participants are not balancing exploitation versus exploration but instead adapting their collection strategies within clearly defined, stable economic yet potentially volatile social context. Furthermore, human participants playing for modest monetary rewards differs from naturalistic animal foraging in terms of ecological stakes. Despite these differences, our paradigm is grounded in fundamental principles of foraging, as participants must dynamically weigh action and opportunity costs, allocate effort and make decisions under uncertainty—which arises not from environmental variability but from partly unpredictable actions of their partner. The sequential nature of decisions under spatiotemporal continuity creates a setting where distinct cooperative and competitive strategies emerge spontaneously, making this an ecologically valid variation of foraging in shared resource environments[86].

## Determinants of payoff efficiency and cost of cooperation

Analyzing the payoff efficiency showed that many participants did not adhere to their individual optimal strategy. Our analysis revealed two variables shaping the payoff function along the cooperation–competition spectrum unique to continuous action spaces: (i) the effect of an increase in movement speed for higher degrees of competition, associated with increased payoffs, and (ii) within-dyad sensorimotor skill differences. These factors are omnipresent in real-world scenarios and other continuous action spaces[31,44], yet they are not captured in discrete (in time or in space) experiments. Beyond a competitive advantage, higher movement speeds towards single targets might reflect higher motivation when foraging for oneself[87–89]. At the same time, our analysis highlights the critical role of skill differences, since they determine the optimal balance between cooperation and competition, both within our task and presumably across other continuous action spaces. But despite the opportunity for more skilled participants to maximize their individual payoff in a selfish manner, many adhered to overly cooperative strategies, paying a cost of cooperation.

The reluctance to exploit more competitive strategies, even when they are optimal, is likely an interplay of multiple factors. It might be explained by prosocial tendencies, characterized by empathy and a preference for fairness and reciprocity[90–92], as well as social norms[93–95]. In addition, once spontaneously established, social conventions may act as a stabilizing force, encouraging participants to adhere to a certain level of cooperation to maintain predictability and successful coordination[28,96,97]. The stabilizing effect of continuous transparent co-action is in line with continuous-time models of cooperation, which show how the propensity of agents to initiate cooperation and to mirror their partner's actions stabilizes at an

evolutionary equilibrium[19]. Cooperation may also be a pragmatic strategy to reduce both physical and cognitive load. As in real foraging or hunting, participants might prioritize strategies that require less rapid, energy-intensive movements, thereby valuing comfort and sustainability over competitive optimization in a long experiment. Many dyads, especially the three fully cooperative turn-taking dyads, alternated between the joint targets, minimizing decision complexity[98]. Lastly, successful cooperation might be less stress-inducing or more rewarding, and thereby subjectively more appealing than competition[99].

Cooperative tendencies in our study might be further amplified by the face-to-face visibility and the transparency of action consequences[19,27]. Even though the actual interaction took place on the abstract game field between the two virtual agents, the presence and direct visibility of a real partner has likely increased social salience[100]. Indeed, it has been demonstrated that dynamic reciprocity is abolished when participants believe that they interact with a computer agent[101]. At the same time, the attribution of agency and intentions to movements of virtual agents[102,103] was, in addition to our results, apparent in the emotional reaction of viewers to gameplay sequences (to experience this firsthand, we encourage readers to view the gameplay videos: Supplementary Movies).

## Understanding continuous interactions

Our CCF game captures the continuous, dynamic nature of realistic social interactions by letting dyads to navigate a shared space. When watching the gameplay movies, the remarkable behavioral complexity of continuous spatiotemporal interactions and mutual coupling between agents become very apparent. Therefore, the transition towards studying continuous decisions requires not only the development of tasks that capture the range of archetypal interactions, but also sophisticated analysis techniques[39,59]. Despite the strategic diversity, we successfully modeled a large part of the observed interactions and characterized the inherent links between target choice uncertainty, trajectories, and the underlying continuous decision processes. The trajectories, however, clearly provide more information about the continuous decision processes than could be captured by our modeling approach. Due to game continuity and action transparency mediating mutually coupled interactions, we observed many high curvature trajectories reflecting miscoordination, indecision, changes of mind or online adjustments to partner's actions, consistent with the integration of information over short timescales[104]. Models that approximate the continuity of the spatiotemporal trajectories more closely, i.e. by predicting actual spatiotemporal dynamics that incorporate ongoing mutual coupling, represent a promising research direction. To make the game even more real-world-like and elicit richer dynamics, future experiments might manipulate the ambiguity of information about the targets and the partner actions, or make the landscape non-uniform in terms of effort and reward probability[16,59]. These adaptations also offer an opportunity to infer the decision points in a large number of interactions, which could serve as salient alignment points for the neural analysis of continuous decisions[44,47,59,105].

## Limitations

Our analysis of continuous decisions along the cooperation–competition spectrum yields valuable insights for the design of future experiments. First, it emphasizes the importance of factoring in the effect of increased movement speed during competitive behavior when formulating the payoff. While we successfully demonstrated that participants performed overly cooperative strategies that incur an associated cost of cooperation, our "flat" payoff formulation did not allow manifesting strategies that are overly competitive and result in a "cost of competition". This is because we did not anticipate, and hence did not compensate for the increase in speed during competition in our payoff formulation. By decreasing the payoff for single targets, we hypothesize that manifestations and quantification of such overly competitive strategies will become possible. The second insight is that the within-dyad skill difference determines the optimal trade-off between cooperation and competition. We developed a post-hoc competitive skill difference measure to estimate the cost of cooperation for intermediate dyads, but its reliability is inherently dependent on the frequency of single target collections, and for highly cooperative dyads (FST < 0.1) we could only estimate a lower bound due to limited data. In future work, an independent calibration of skill closely aligned with the task should provide a cleaner dissociation between strategic choices and individual motor or planning abilities, and a more robust and comprehensive estimation of the cost of cooperation.

Finally, while we can reliably predict stable decision-making, it remains unknown what causes the broad strategic diversity in the first place. This raises the question of whether the convergence to a specific strategy in each dyad is a consequence of (i) few initial spatial configurations that strongly shape the ensuing interactions (akin to a complex system evolving towards an attractor), (ii) a specific combination of personality predispositions of the two participants[106–109], or (iii) an interplay of both factors. Future research should explore these possibilities by systematically manipulating initial conditions, assessing personality traits, and leveraging computational modeling to disentangle their relative contributions to strategic convergence.

## Summary

In summary, we contribute to the growing field of continuous decisions by developing a richly flexible but tractable paradigm that affords cooperative and cooperative strategies within the same social context, under conditions of direct action visibility. Our analysis reveals the spontaneous emergence of stable strategies, spanning from cooperation to intermediate strategies to pure competition. The model incorporating weighted path minimization and across-cycle dynamics demonstrates that dyads with intermediate strategy rely on sensorimotor communication to facilitate coordination between cooperation and competition. We show that preceding interactions together with the initial conditions at the start of each collection cycle shape the decision uncertainty and within-cycle spatiotemporal dynamics, ultimately giving rise to specific target choices. These results form a solid basis for future research aimed at identifying where and how these factors are represented in the brain and exploring the interplay between different strategies, individual personality traits, and social contexts.

## Data availability

The data for this study are available at the public Open Science Framework repository, https://osf.io/56hw7, https://doi.org/10.17605/osf.io/56hw7.

## Code availability

The code related to this study, including the code to run the experiment, analyses and simulations, is available at the public Open Science Framework repository, https://osf.io/56hw7[110], https://doi.org/10.17605/osf.io/56hw7.

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

## Acknowledgements

We thank Dr. Sebastian Moeller and Klaus Heisig for the technical support with the Dyadic Interaction Platform, Dr. Annika Ziereis for helpful comments on the manuscript, Mariia Kadochnikova and Dr. Zahra Yousefi Darani for taking part in the game demonstration movie, and Karin Tilch and Thorge Beilfuß for the photos and the game demonstration movie, respectively. We thank Dr. Chris Schloegl for efficient scientific coordination of the Leibniz ScienceCampus Primate Cognition and the Collaborative Research Center SFB 1528 "Cognition of Interaction", and the members of these consortium for stimulating discussions. This work was supported by the Leibniz ScienceCampus Primate Cognition (to AG and IK), the Leibniz Collaborative Excellence grant K265/2019 "Neurophysiological mechanisms of primate interactions in dynamic sensorimotor settings" (to AG and IK), the Cluster of Excellence "Multiscale Bioimaging" (MBExC, to VP), and the German Research Foundation via the Collaborative Research Center "Cognition of Interaction" (DFG SFB-1528, projects A06, C02 and Z01, to LP, AS, AG, VP, and IK). The funders had no role in study design, data collection, and analysis, decision to publish, or preparation of the manuscript.

## Author contributions

Conceptualization: D.L., V.I., J.D., V.P., I.K. Data curation: D.L., V.I. Formal analysis: D.L., J.D. Funding acquisition: L.P., A.S., A.G., V.P., I.K. Investigation: D.L., V.I., J.R., A.F. Methodology: D.L., V.I., V.P., I.K. Project administration: L.P., A.S., A.G., V.P., I.K. Software: D.L. Resources: D.L., V.I., V.P., I.K. Supervision: J.D., V.P., I.K. Validation: D.L. Visualization: D.L., V.I., J.D., V.P., I.K. Writing—original draft: D.L., J.D., V.P., I.K. Writing—review and editing: D.L., V.I., J.D., A.G., V.P., I.K.

## Funding

## Competing interests

The authors declare no competing interests.
