## [Transparent Peer Review file · Communications Psychology]

Continuous dynamics of cooperation and competition in social decision-making

Corresponding Author: Dr Igor Kagan

Version 0:

Decision Letter:

Dear Dr Kagan,

Thank you for your patience during the peer-review process. Your manuscript titled "Continuous dynamics of cooperation and competition in social foraging" has now been seen by 3 reviewers, and I include their comments at the end of this message. They find your work of interest but raised some important points. We are interested in the possibility of publishing your study in Communications Psychology, but would like to consider your responses to these concerns and assess a revised manuscript before we make a final decision on publication.

We therefore invite you to revise and resubmit your manuscript, along with a point-by-point response to the reviewers. Please highlight all changes in the manuscript text file.

Editorially, we ask you to carefully address the methodological and conceptual concerns raised by R1 and 2 (e.g., details about how skill of players to play the game was measured). Please also consider re-organizing the results part to improve the clarity, as suggested by R3.

I am attaching an Editorial Requests Table that details critical reporting requirements for the revised manuscript. Please attend to each item and ensure your manuscript is fully compliant. If your revised manuscript is not aligned with these requests on major issues, such as those concerning statistics, it may be returned to you for further revisions without re-review.

Please submit the following items:

- Revised manuscript
- Point-by-point response to the referees' comments
- Cover letter (as a separate document)
- <https://www.nature.com/documents/nr-reporting-summary.pdf> Nature Research Reporting Summary
- Completed Editorial Request Table (attached).

via this link: Link Redacted .

Additional guidance is available in our style and formatting guide Communications Psychology formatting guide.

Best regards,

Troy Lui, on behalf of

Yafeng Pan

Troy Lui, PhD
Associate Editor
Communications Psychology

Yafeng Pan, PhD
Editorial Board Member
Communications Psychology
orcid.org/0000-0002-5633-8313

REVIEWER EXPERTISE:

Reviewer #1: cooperation and competition, social decision-making, computational modeling
Reviewer #2: cooperation and competition, social decision-making, computational modeling
Reviewer #3: cooperation and competition, social decision-making, computational modeling

REVIEWER REPORTS:

Reviewer #1 (Remarks to the Author):

The manuscript Continuous dynamics of cooperation and competition in social foraging considered for submission in Communications psychology introduces a novel experimental paradigm based on a game played in pairs facing each others and interacting via a transparent screen separating the two players. The task consists in developing strategies, either competing with the other player or cooperating with them, to get financial rewards in a spatially-explicit continuous environment. The manuscript presents the paradigm, an experiment made of 2x20 min trials from 62 different pairs and reports a detailed and thorough analysis of the collected data.

Overall, the manuscript is very well written, figures (including captions) of high quality and the provided analysis exhaustive and in-depth. The reported results support well the motivations of studying this novel experimental paradigm further. I particularly appreciated that in addition to the novel experimental paradigm, many in-depth ideas about how to analyse the data coming from the experiments are already provided.

My main concern regards how skill of players to play the game is measured. I am not fully convinced that what has been measured by the authors actually reflects skills of players and improves the understanding of the system. Skill in the manuscript has been defined as the difference in single targets collected, normalized by the total number of single targets. This choice was motivated by the finding that the correlation between payoff difference and single target difference was very high. This correlation is however heavily biased since trivially perfect for pairs with high FSTs, which, by definition, are only discriminated by single target differences. Also, by construction, this definition excludes pairs that chose to avoid the single targets in their strategies. Therefore, this metrics seems biased in its explanatory power in favour of high FSTs, cannot encompass all the strategies exhibited by players and may not be fully linked to skills of players. It seems, for instance, that strategies themselves may lead the players to be in very different mindsets – highly coordinated strategies in which players know with high certainty how the other will play (either focusing on joint targets or dividing the arena in two) presumably leading players to play more automatically with a smaller cognitive load (as suggested by the authors L553-554). In that case, the difference seen in payoff may be more a disinterest in the actual final payoff than clear skill differences.

I am also sceptical about the reference to foraging theory, especially in the title of the manuscript: it is indeed interesting to draw connections with behavioural ecology and animal decision-making and to discuss those but I am not sure the game primarily intends to investigate the questions associated with these fields. This paradigm surely allows to investigate human decision-making in the context of a game with small impacts on the lives of players but I find the connection to animal foraging to be a fairly bold claim.

Minor comments:

L192: There is no subpanel in Figure S2, S2a is not required

Fig 3d, caption: maybe adding (b) to the sentence "The actual dyads lie between simulated strategies" would make clearer that the two lines are the same ones as on panel b.

Fig 4a, caption: what is 'invite placements'? This is not explained yet when we first get the figure introduced (L212) – explanation only comes L231.

L388 Although the statement of this conclusive paragraph (that cooperative dyads get lower joint payoffs) is in general true on Fig 6a, it could be slightly nuanced by what the authors reported above: when cooperation is highly coordinated (diamonds), they result in higher payoffs compared to dyads with similar FSTs and the actual best dyad overall could also be described as cooperation (the blue + symbol) since the two players divided the space equally and actively avoid competition.

L651: rolled a die > a dice

Reviewer #2 (Remarks to the Author):

In the manuscript entitled "Continuous dynamics of cooperation and competition in social foraging" the authors propose a new game, the Cooperation-Competition-Foraging game, to study social interactions of dyads navigating a continuous shared space while foraging for rewards. Members of the dyads can collect rewards either together or independently.

The task involves a free-flowing social game, where two participants use continuous arm movements to collect targets in a face-to-face setting with full transparency about their co-player movements and appearance. The task consists of moving a mouse on a 2D screen towards either one of two cooperative targets to be reached together with the other player or towards an individual target to be reached before the other player.

Subjects played for two blocks of 20 minutes each and after the first 10-15 minutes most dyads' behaviours conformed to stable strategies with groups of dyads opting for either mainly cooperative, mainly competitive or intermediate strategies as estimated by the fraction of single targets reached over a short period of interactions.

The authors derive optimal strategies based on principles of path minimisations and cooperative strategies to achieve it and show that a theoretically optimal fraction of single targets (FST) stands between 0.33 and 0.4 depending on whether participants move together or spread themselves in the space following single targets.

They develop a model based on these principles and on observed behavioural patterns of prioritising targets not chosen on the previous trial and prioritising cooperation after competition. They further characterise classes of behaviour and the spatio-temporal features underlying them as well as the impact of difference in skills in the game.

Overall this is a really nice paper, building on a fantastic new task to study the trade off between cooperation and competition, a multifaceted, engaging and dynamic social foraging game on 2D that taps in the continuous nature of social interactions as well as the sensorimotor behaviour that underlies them. The game provides for an extremely rich repertoire of social behaviours which is both well modelled and well described by a compelling set of analysis and a commendable collection of videos. I congratulate the authors for their work and I am really glad if they found any inspiration on our earlier work on the Space Dilemma.

My concerns are minor and are mostly aimed at increasing clarity and understanding for the reader.

My most significant clarification relates to the definition of the most advantageous position. How the authors derive the advantageous position in equation 11 and 12 is theoretically sound but I found the colormap in figure 3c is somewhat confusing. I understand the map should identify the most advantageous position to minimise the distance to the next target irrespective of whether this is the single target or the joint target closest to the collecting agent. So it follows that, intuitively, it should be the combination of two maps, one relating to the set of best positions to get to the closest joint target, and one to the position that minimises the dyad distance to the new individual target, each weighted by the relative target type weighting parameter. The former map would be a circle centered around the joint target with radius equal to the distance from the agent who collected the previous round (like in sup fig.1b). The latter map would look something like sup fig1 a, with the best position being symmetrical with respect to the midpoint from the collecting agent. However in fig.3c it seems like the gradient mainly reflects the distance from the closest joint target. Can you provide an intuition both in the legend and text of why that is instead of a blob around the advantageous position that combines the two maps? Is it because in that example

the weight is highly biased towards joint collection? In general, the weight will make a difference on what this advantageous position heatmap look like and it would be good if that was articulated in the text and perhaps with two different example heatmaps for different weights.

Relatedly in figure S1 b, it's counterintuitive why position closer to the closest joint target should be preferable to positions equidistant from the closest joint but further away from the collecting agent (as symmetrical from the midpoint as possible). My understanding is that if only the maximum distance to the closest joint matters there is no point for the non collecting agent in being closer to it than the collecting agent. Could the authors perhaps show the two maps representing the maximum distance to the closest joint target and the minimum average distance to the unknown individual target both separately and combined? This would potentially make those maps more intuitive. At the moment I can't get my head round why S1 b is qualitatively different than S1c

In figure 3a the weights for the distance are not quite clear. Are they represented visually somewhere? From the legend it seems only the weighted distances are. So is the orange weighted distance always higher than the blue one because the original distances are? That's not immediately apparent from the figure. The "joint preferred" condition with three different weighted distances can be confusing. Could you also somehow visually or numerically represent the weights? Or report the numbers in the legend? That would help understand more intuitively why on the left joint wins and on the right the individual target. Perhaps also highlight the winning target in each condition.

Likewise in figure 3b it would also help to get a numerical sense of the three weights conditions, in the legend or in the panel.

Can the authors swap panel 3b and 3c? Conceptually it makes sense to explain graphically the advantageous position before the simulation and current panel b and d could look nicer side by side?

Caption figure 3d most dyads don't seem to be in between simulated strategies? Not clear what the red lines measure. Is it the distance from the "always same starting position" curve (That is, what reveals their degree of adv starting position)? It doesn't look so from the plot. Why?

Have the authors plan to modulate behaviour by having different rewards structures/conditions in future experiment and can the authors comment in the discussion what happens if one eliminates the collaborative targets? It seems to me that would provide a 2D generalisation of the intermediate condition in the Space Dilemma.

Can the authors comment on whether the reduction in the shift towards the advantageous position depends on the risk of increasing the chance of the co-player winning the next single trial and therefore can be framed as a matter of trust?

Minor points

The diamond dyads do the turn-take which is interesting. From the video it looks like they do it almost normatively, irrespective of distance of the other two targets. Is that so or there are exceptions for the situations where those other targets are closer?

Even though the sample was highly biased, did you observe any gender differences?

Reviewer #3 (Remarks to the Author):

This study marks a novel approach in the fields of both game theory and foraging. They use a "transparent game" method to study continuous dynamics of cooperation and competition in a foraging setting. I really like their approach and find it to be an important contribution to multiple fields, in terms of both the unique findings and the methodological approach. However, I have some suggestions for improvement regarding readability of the paper:

1. The Introduction is comprehensive but it is quite lengthy and redundant at times. I would suggest making it more concise. It also focuses a lot on the methodological advancements, which is warranted, but it seems to be the only focus. In a way I was unprepared for some elements of the task design (such as single and joint targets) and of the results ("invitation", different strategies, skill difference etc.). I think the authors can foreshadow these concepts in the introduction and motivate these aspects of their design, in addition to current focus on continuous, face-to-face dynamics.

2. I found that the results are also quite verbose and hard to follow at some times because of the high amount of information. Maybe the authors can move some plots/sub-results to the supplementary to make it easier for a reader to grasp the results. It also would be helpful to end the introduction or start the results by explaining the different strategies and what are their signatures.

3. Line 128 - "transporting" reads weird.

4. In one of the results subheadings, there is "shaping" -- the authors probably mean "shapes/shape".

Version 1:

Decision Letter:

Dear Dr Kagan,

Your manuscript titled "Continuous dynamics of cooperation and competition in social decision-making" has now been seen by our reviewers, whose comments appear below. In light of their advice I am delighted to say that we are happy, in principle, to publish a suitably revised version in Communications Psychology.

We therefore invite you to revise your paper one last time to address the remaining concerns of our reviewers and a list of editorial requests. At the same time we ask that you edit your manuscript to comply with our format requirements and to maximise the accessibility and therefore the impact of your work.

EDITORIAL REQUESTS:

SUBMISSION INFORMATION:

OPEN ACCESS:

* **DATA AVAILABILITY:**

Link Redacted

Best regards,

Troy Lui, on behalf of

Yafeng Pan

Troy Lui, PhD
Associate Editor
Communications Psychology

Yafeng Pan, PhD
Editorial Board Member
Communications Psychology
orcid.org/0000-0002-5633-8313

REVIEWERS' COMMENTS:

Reviewer #1 (Remarks to the Author):

The authors appropriately addressed all my comments.

I spotted one typo:
L398: 'hightest' > highest

Reviewer #2 (Remarks to the Author):

I am happy with the authors' revisions and for the paper to be published.

Reviewer #3 (Remarks to the Author):

I thank the authors for their revisions and for this work! I have no further comments.

MAX PLANCK INSTITUTE
FOR DYNAMICS AND
SELF-ORGANIZATION

Deutsches Primatenzentrum
Leibniz-Institut für Primatenforschung

Prof. Viola Priesemann
Max Planck Institute for Dynamics and Self-Organization
Am Fassberg 17, 37077, Göttingen
viola.priesemann@ds.mpg.de

Dr. Igor Kagan
German Primate Center – Leibniz Institute for Primate Research
Kellnerweg 4, 37077, Göttingen
ikagan@dpz.eu

July 17, 2025

Troby Lui, PhD
Associate Editor

Yafeng Pan, PhD
Editorial Board Member

Response to Reviewers

Dear Dr. Lui, dear Dr. Pan, dear Reviewers,

We are grateful for the opportunity to submit the revised manuscript, which incorporates the revisions requested by the three Reviewers. We would like to express our appreciation for the very fast and positive evaluation of our work, and for thoughtful and constructive questions and suggestions that significantly improved the manuscript.

To summarize our revisions, we carefully addressed the methodological and conceptual questions raised by the Reviewers, such as providing more details about skill metric, and clarifying advantageous placement and target type weighting. We also made the Introduction more concise, improved a lead-up to key concepts, and streamlined the Results to improve the clarity.

Below we provide a point-by-point reply to the Reviewers' comments. The comments are in *italics*; our responses are in the regular font.

In the revised article file with tracked changes, the substantial changes are highlighted in light blue. We refrained from copying the revised text to this response for brevity, and use the line numbers (L...) to indicate the location of specific changes in response to the comments.

Reviewer 1

[...] Overall, the manuscript is very well written, figures (including captions) of high quality and the provided analysis exhaustive and in-depth. The reported results support well the motivations of studying this novel experimental paradigm further. I particularly appreciated that in addition to the novel experimental paradigm, many in-depth ideas about how to analyse the data coming from the experiments are already provided.

We thank Reviewer 1 for the positive evaluation and the appreciation of our analysis approaches.

My main concern regards how skill of players to play the game is measured. I am not fully convinced that what has been measured by the authors actually reflects skills of players and improves the understanding of the system. Skill in the manuscript has been defined as the difference in single targets collected, normalized by the total number of single targets. This choice was motivated by the finding that the correlation between payoff difference and single target difference was very high. This correlation is however heavily biased since trivially perfect for pairs with high FSTs, which, by definition, are only discriminated by single target differences. Also, by construction, this definition excludes pairs that chose to avoid the single targets in their strategies. Therefore, this metrics seems biased in its explanatory power in favour of high FSTs, cannot encompass all the strategies exhibited by players and may not be fully linked to skills of players. It seems, for instance, that strategies themselves may lead the players to be in very different mindsets – highly coordinated strategies in which players know with high certainty how the other will play (either focusing on joint targets or dividing the arena in two) presumably leading players to play more automatically with a smaller cognitive load (as suggested by the authors L553-554). In that case, the difference seen in payoff may be more a disinterest in the actual final payoff than clear skill differences.

We agree with the Reviewer's concern and appreciate the opportunity to clarify and improve this aspect of our manuscript. First, we note that the measure of skill based on single target collection was calculated only for collection cycles in which **both** participants actively aimed for single targets (as we now clearly state in the Results, L412-420, and the revised Fig. 7 caption, as well as in the Methods, L759-765). We acknowledge that this was not emphasized clearly enough in the Results and the figure caption, and our previous explanation in the Methods section was confusing, which might have contributed to a misunderstanding of this metric.

Second, even if imperfect, this approach provides a way to quantify an important aspect of social behavior — namely, the cost of cooperation, particularly for the more skilled player who could have, counterfactually, adhered to a more competitive strategy to maximize individual payoff. We contend that regardless of the underlying factors driving less successful collection of single target by one of the players — be it disinterest in the payoff or actual sensorimotor ability — the measure still reflects a relative advantage, motivational or sensorimotor, of the better-performing player in competitive contexts.

That said, we fully recognize that this measure of skill is limited: its reliability is inherently dependent on the frequency of single target collections, it cannot be estimated in dyads who avoided single targets entirely, and it does not capture cooperative skill in dyads with stable cooperative tactics (e.g., field division or alternating turns). Additionally, as the reviewer points out, strategic choice — particularly in highly coordinated dyads — can reduce cognitive load and make performance less

reflective of individual competitive sensorimotor abilities. These factors limit the generality of our current metric.

We have revised the manuscript to make these limitations explicit. We now introduce it as “**competitive skill**”, and explain the underlying assumptions: i) if both agents move straight to the single target, they compete, and ii) the competitive skill is independent of FST (L412-420, and L437-438). In the Limitations section of the Discussion (L619-626), we now describe this measure as a post-hoc and potentially biased proxy for competitive skill. We also emphasize that a more robust and generalizable assessment might require an independent skill calibration block, in which participants are explicitly instructed to compete for single targets under controlled conditions. Such a manipulation would allow for a cleaner dissociation between strategic choices and individual motor or planning abilities. While our current data do not permit this, we present the analysis as a useful step toward understanding the relationship between competitive skill differences, payoff inequality, and cost of cooperation, within certain strategic regimes.

I am also sceptical about the reference to foraging theory, especially in the title of the manuscript: it is indeed interesting to draw connections with behavioural ecology and animal decision-making and to discuss those but I am not sure the game primarily intends to investigate the questions associated with these fields. This paradigm surely allows to investigate human decision-making in the context of a game with small impacts on the lives of players but I find the connection to animal foraging to be a fairly bold claim.

We appreciate the reviewer’s critique and agree that our paradigm differs from naturalistic animal foraging in terms of ecological stakes. We have changed the title to “Continuous dynamics of cooperation and competition in social **decision-making**” to emphasize the decision-making framework while maintaining the key features of dynamic interaction.

Nonetheless, we respectfully argue that core principles from foraging theory remain highly relevant. Even in laboratory settings with modest rewards, the foraging framework provides a well-established and generalizable approach to studying cost-benefit trade-offs, spatiotemporal choices, and strategy adaptation under uncertainty — all of which are present in our task. Indeed, a substantial body of human and primate neuroscience research has successfully applied foraging theory to analyze stay/leave decisions, reward history effects, information sampling, and exploration–exploitation dynamics under similar low-stakes conditions (e.g. [1-6]). Likewise, our game instantiates a continuous decision space in which participants must weigh action costs, opportunity costs, and the predicted behavior of a social partner across a sequence of choices — hallmarks of foraging-like decisions. We have added a clarification to the Discussion to articulate this reasoning more clearly (L543-551).

[1] Kolling, N., Behrens, T. E. J., Mars, R. B., Rushworth, M. F. S. (2012). Neural mechanisms of foraging. *Science*, 336(6077), 95–98.

[2] Hayden, B. Y., Pearson, J. M., Platt, M. L. (2011). Neuronal basis of sequential foraging decisions in a patchy environment. *Nature Neuroscience*, 14(7), 933–939.

[3] Pearson, J. M., Hayden, B. Y., Raghavachari, S., Platt, M. L. (2009). Neurons in posterior cingulate cortex signal exploratory decisions in a dynamic multioption choice task. *Current Biology*, 19(18), 1532–1537.

[4] Hall-McMaster, S., Luyckx, F. (2019). Revisiting foraging approaches in neuroscience. *Cognitive, Affective, Behavioral Neuroscience*, 19(2), 225–230.

[5] Gabay, A. S., Apps, M. A. J. (2021). Foraging optimally in social neuroscience: computations and methodological considerations. *Social Cognitive and Affective Neuroscience*, 16(8), 782–794.

[6] Mobbs, D., Trimmer, P. C., Blumstein, D. T., Dayan, P. (2018). Foraging for foundations in decision neuroscience: insights from ethology. *Nature Reviews Neuroscience*, 19(7), 419–427.

Minor comments:

L192: There is no subpanel in Figure S2, S2a is not required

Thank you for noticing the inconsistency, done.

Fig 3d, caption: maybe adding (b) to the sentence “The actual dyads lie between simulated strategies” would make clearer that the two lines are the same ones as on panel b.

Thank you, we extensively revised this caption (also in response to Reviewer 2), please see page 7.

Fig 4a, caption: what is ‘invite placements’? This is not explained yet when we first get the figure introduced (L212) – explanation only comes L231.

We placed this figure after the text where invitations are first introduced.

L388 Although the statement of this conclusive paragraph (that cooperative dyads get lower joint payoffs) is in general true on Fig 6a, it could be slightly nuanced by what the authors reported above: when cooperation is highly coordinated (diamonds), they result in higher payoffs compared to dyads with similar FSTs and the actual best dyad overall could also be described as cooperation (the blue + symbol) since the two players divided the space equally and actively avoid competition.

Thank you, we added these points (L393-401).

L651: rolled a die > a dice

Respectfully, die is singular, dice is plural (the form “rolled the dice” might be more familiar because dice are frequently used in pairs in games).

Reviewer 2

[...] Overall this is a really nice paper, building on a fantastic new task to study the trade off between cooperation and competition, a multifaceted, engaging and dynamic social foraging game on 2D that taps in the continuous nature of social interactions as well as the sensorimotor behaviour that underlies them. The game provides for an extremely rich repertoire of social behaviours which is both well modelled and well described by a compelling set of analysis and a commendable collection of videos. I congratulate the authors for their work and I am really glad if they found any inspiration on our earlier work on the Space Dilemma.

We thank Reviewer 2 for very positive and in-depth evaluation, and for their earlier work on the

Space Dilemma that indeed shaped much of our thinking and interpretations.

My concerns are minor and are mostly aimed at increasing clarity and understanding for the reader.

My most significant clarification relates to the definition of the most advantageous position. How the authors derive the advantageous position in equation 11 and 12 is theoretically sound but I found the colormap in figure 3c is somewhat confusing. I understand the map should identify the most advantageous position to minimise the distance to the next target irrespective of whether this is the single target or the joint target closest to the collecting agent. So it follows that, intuitively, it should be the combination of two maps, one relating to the set of best positions to get to the closest joint target, and one to the position that minimises the dyad distance to the new individual target, each weighted by the relative target type weighting parameter. The former map would be a circle centered around the joint target with radius equal to the distance from the agent who collected the previous round (like in sup fig.1b). The latter map would look something like sup fig1a, with the best position being symmetrical with respect to the midpoint from the collecting agent. However in fig.3c it seems like the gradient mainly reflects the distance from the closest joint target. Can you provide an intuition both in the legend and text of why that is instead of a blob around the advantageous position that combines the two maps? Is it because in that example the weight is highly biased towards joint collection? In general, the weight will make a difference on what this advantageous position heatmap look like and it would be good if that was articulated in the text and perhaps with two different example heatmaps for different weights.

Relatedly in figure S1b, it's counterintuitive why position closer to the closest joint target should be preferable to positions equidistant from the closest joint but further away from the collecting agent (as symmetrical from the midpoint as possible). My understanding is that if only the maximum distance to the closest joint matters there is no point for the non collecting agent in being closer to it than the collecting agent. Could the authors perhaps show the two maps representing the maximum distance to the closest joint target and the minimum average distance to the unknown individual target both separately and combined? This would potentially make those maps more intuitive. At the moment I can't get my head round why S1b is qualitatively different than S1c.

We appreciate the thoughtful and detailed nature of these inquiries, which highlighted a lack of clarity in our previous presentation, for which we apologize. The reviewer's intuition and understanding are correct. The panel in the original Fig. 3c (now 3b) — the same panel reproduced in Fig. S1c — is already a weighted combination of the two maps: (i) the principle demonstrated in Fig. S1a (“ignore joint targets; minimize distance to single target”), and (ii) the principle minimizing the distance to the closest joint target. Both panels S1b and S1c represent the combination of these principles, but the relative contribution of single and joint targets depends on the distance to the closest joint target, and the single/joint target weighting. Due to the fixed color scale and saturation of the “joint target map” in S1b, the contribution of the single target map (as in S1a) was not very apparent.

We now replotted these panels with a more intuitively oriented and better scaled color bar that clearly indicates the cutoff below 6.5 cm, added explanations and arrows pointing to the optimal placement(s) to each panel, and extensively revised Figure S1 and Fig. 3 captions. We also explicitly indicate the weighting factor ($w=0.5$ in Fig. S1 panels (a-d), $w=0.99$ in (e)). We also updated the Results (L165-166).

In figure 3a the weights for the distance are not quite clear. Are they represented visually somewhere?

From the legend it seems only the weighted distances are. So is the orange weighted distance always higher than the blue one because the original distances are? That's not immediately apparent from the figure. The "joint preferred" condition with three different weighted distances can be confusing. Could you also somehow visually or numerically represent the weights? Or report the numbers in the legend? That would help understand more intuitively why on the left joint wins and on the right the individual target. Perhaps also highlight the winning target in each condition. Likewise in figure 3b it would also help to get a numerical sense of the three weights conditions, in the legend or in the panel.

Good points; we added the actual weights and also indicate which target is collected with an asterisk. Yes, in the examples shown, the orange weighted distance is always higher than the blue one because of the original distances. We now mention that the weights in previous panel 3b (now 3c) are the same as in 3a.

Can the authors swap panel 3b and 3c? Conceptually it makes sense to explain graphically the advantageous position before the simulation and current panel b and d could look nicer side by side?

Great suggestion, we swapped the panels.

Caption figure 3d most dyads don't seem to be in between simulated strategies? Not clear what the red lines measure. Is it the distance from the "always same starting position" curve (That is, what reveals their degree of adv starting position)? It doesn't look so from the plot. Why?

Yes, the red lines indicate the distance reduction due to some degree of advantageous placement, i.e. the distance reduction due to the free agent not sharing the same starting positions with the collecting agent, as in the simulations that yield the "always same starting position" curves. For the $FST=1$ dyads, the tops of the red lines do not perfectly align with the dark gray dotted curve because the simulations reflect an idealized approximation over a very large number of target collections, while the dyads' actual means are based on finite, variable data and therefore include some jitter. We clarified these points in the legend in Fig. 3d, and in the extensively revised caption.

Have the authors plan to modulate behaviour by having different rewards structures/conditions in future experiment and can the authors comment in the discussion what happens if one eliminates the collaborative targets? It seems to me that would provide a 2D generalization of the intermediate condition in the Space Dilemma.

Yes, we plan to build on this game to investigate different aspects of human and macaque behavior, including manipulations of payoff structure — similarly to what has been done in the Space Dilemma study in PISAURO et al. 2022 — and information availability. But our currently ongoing project uses the same formulation in a round-robin design, investigating the relationship between personality and hormonal traits and convergence to a specific strategy, and neural correlates of those strategies. Reviewer 2 is absolutely correct that the CCF game without the joint targets would be a generalization of the Space Dilemma in 2D, adding the target collection movements and mutually observed real-time actions and thus, continuity not only in space but also in time.

Can the authors comment on whether the reduction in the shift towards the advantageous position depends on the risk of increasing the chance of the co-player winning the next single trial and therefore can be framed as a matter of trust?

Yes, the (non-competitive) advantageous placement can indeed increase the co-player's chance of winning the next single target, and thus can be framed as a potential indicator of trust that other one will do the same. However, interpreting it as a direct measure of trust is challenging. In an idealized scenario — with no skill difference and instantaneous reaction times — the competitive placement would be right next to the collecting agent, and any extra distance towards the center could be attributed to trust. In practice, however, this strategy might not be very obvious to the naive participants, and real players might try to gain a competitive **timing** advantage by shifting toward the center. Therefore, this positioning can reflect either trust (non-competitive, advantageous placement for either player) or competitive heuristics based on perceived skill and reaction times, making it difficult to disentangle the two without incorporating actual movement skill and reaction times into a model.

We note that the invitations to the joint targets constitute a more reliable measure of trust. We now mention both of these aspects in the Discussion (L505 and the next paragraph, L506-513).

Minor points

The diamond dyads do the turn-take which is interesting. From the video it looks like they do it almost normatively, irrespective of distance of the other two targets. Is that so or there are exceptions for the situations where those other targets are closer?

Indeed, these dyads ignored the distance and performed strict normative turn-taking — we now emphasize it more in L279-282.

Even though the sample was highly biased, did you observe any gender differences?

It is undoubtedly very interesting question but beyond the scope of our study due to the sample limitations, as appreciated by the Reviewer. Anecdotally, few female and mixed dyads we tested (7 female dyads, 5 mixed female/male dyads) exhibited a similar distribution of FST values, from 0 to 1, showing cooperative, intermediate and competitive strategies.

Reviewer 3

This study marks a novel approach in the fields of both game theory and foraging. They use a "transparent game" method to study continuous dynamics of cooperation and competition in a foraging setting. I really like their approach and find it to be an important contribution to multiple fields, in terms of both the unique findings and the methodological approach. However, I have some suggestions for improvement regarding readability of the paper:

We thank Reviewer 3 for the positive feedback and for recognizing the important contributions of our work. We appreciate these suggestions and have revised the manuscript to improve its readability.

1. The Introduction is comprehensive but it is quite lengthy and redundant at times. I would suggest making it more concise. It also focuses a lot on the methodological advancements, which is warranted, but it seems to be the only focus. In a way I was unprepared for some elements of the task design (such as single and joint targets) and of the results ("invitation", different strategies, skill difference etc.). I think the authors can foreshadow these concepts in the introduction and motivate these aspects of their

design, in addition to current focus on continuous, face-to-face dynamics.

We made the Introduction more concise but also expanded the lead-up to the key design concepts (L72-82).

2. I found that the results are also quite verbose and hard to follow at some times because of the high amount of information. Maybe the authors can move some plots/sub-results to the supplementary to make it easier for a reader to grasp the results. It also would be helpful to end the introduction or start the results by explaining the different strategies and what are their signatures.

We systematically streamlined the narrative in the Results, and added an overview of strategies right after we introduce the three main groups (L139-142). We retained most figures because we felt strongly that they are necessary to present a logical story, but we moved the panels (d), (e) and (f) from Fig. 6, and the corresponding text, to the new Suppl. Fig. S5.

3. Line 128 - "transporting" reads weird.

Thank you, we changed the sentence to: "By embedding foraging in a shared virtual space and a salient social context, our setup provides a controlled yet dynamic environment to study how dyads develop cooperative or competitive strategies."

4. In one of the results subheadings, there is "shaping" – the authors probably mean "shapes/shape".

We changed it to "shape".

Once again, we thank all Reviewers for their time and effort!

Sincerely,

Darius Lewen, Viola Priesemann and Igor Kagan, on behalf of all authors